

# Aerosol composition, air quality, and boundary layer dynamics in the urban background of Stuttgart in winter

Hengheng Zhang[1,6], Wei Huang[1,2], Xiaoli Shen[1,3], Ramakrishna Ramisetty[1,4], Junwei Song[1],
Olga Kiseleva[1], Christopher Claus Holst[1], Basit Khan[1,5], Thomas Leisner[1], and Thomas Saathoff[1]

[1]Institute of Meteorology and Climate Research, Karlsruhe Institute of Technology, Eggenstein-Leopoldshafen, Karlsruhe, Germany
[2]Now at: Institute for Atmospheric and Earth System Research / Physics, Faculty of Science, University of Helsinki, Helsinki, Finland
[3]Now at: Department of Earth, Atmospheric, and Planetary Sciences, Purdue University, 47907, West Lafayette, Indiana, United States
[4]Now at: TSI Instruments India Private Limited, Bangalore, 560102, India
[5]Now at: Mubadala Arabian Center for Climate and Environmental Sciences (ACCESS), New York University, Abu Dhabi, United Arab Emirates
[6]Now at: Research Institute for Applied Mechanics, Kyushu University, Fukuoka, Japan

**Correspondence:** Hengheng Zhang (hengheng.zhang@kit.edu) and Harald Saathoff (harald.saathoff@kit.edu)

**Abstract.** Aerosol distribution is of great relevance for air quality especially for the cities like Stuttgart which is located in a basin. To understand the impact of boundary layer mixing processes on local air quality and to validate the large eddy simulation (LES) model PAML-4U, we collected a comprehensive set of data from remote sensing, *in-situ* methods including radiosondes for the urban background of downtown Stuttgart. Stagnant meteorological conditions caused accumulation of

aerosols during winter. Case studies show that clouds during previous nights can weaken temperature inversion and accelerate boundary layer mixing after sunrise. This is important for ground-level aerosol dilution during morning rush hours. Furthermore, our observations validate results of the LES model PALM-4U in terms of boundary layer heights and aerosol mixing for 48 hours. The simulated aerosol concentrations follow the trend of our observations but are still underestimated by a factor of 4.5 ± 2.1. This underestimation is mainly due to uncertainties of emissions and boundary conditions of the model. This

paper firstly evaluates the PALM-4U model performance in simulating aerosol temporal-spatial distribution, which can help to improve the LES model and to better understand sources and sinks for air pollution as well as the role of horizontal and vertical transport.

## 1   Introduction

Global and regional distribution of aerosol particles are of great concern, partly because they are much more visible than

gaseous pollution (Chan and Yao, 2008; Guo et al., 2009; Li et al., 2016), and because they have discernible adverse effects on human health (Pöschl, 2005; Shiraiwa et al., 2017). Moreover, airborne particles critically impact Earth's climate through aerosol direct and indirect effects (Kiehl and Briegleb, 1993; Ramanathan et al., 2001; Ackerman et al., 2004; Stocker, 2014; Guo et al., 2017). The regional air quality is greatly affected by the temporal and spatial distribution of aerosol within the





planetary boundary layer (Stanier et al., 2004; Li et al., 2017), which are related to the emission of aerosol particles, boundary

layer structures, and the meteorological conditions.

The planetary boundary layer (PBL) is typically about 1 - 2 km thick (10 %-20 % of the troposphere) from the ground surface but can vary from 10 m to 4 km or more (Stull, 1988). Hence, the boundary layer plays a crucial role in supporting human and ecological environments. On larger scales, the PBL greatly affects the whole atmospheric system and determines the exchange of heat, moisture and momentum between the earth's surface and the free troposphere (Garratt, 1994; Medeiros et al., 2005).

The top of the PBL, typically referred to as boundary layer height, marks the transition from the layer of thorough mixing due to turbulence to the free troposphere where mixing is comparatively small. The fundamental definition of the PBL has traditionally been turbulence based. If the mixing is induced by convection it is also called convective boundary layer (CBL) and during night time it is referred to as nocturnal boundary layer (NBL), or stable boundary layer (SBL). The boundary layer is a turbulent layer adjacent to the earth's surface layer (Stull, 1988).

Many methods have been used to investigate the atmospheric parameters (e.g wind and temperature) and constituents (e.g. water and particles) within the PBL. *In-situ* measurements, such as weather sensors deployed at ground level meteorological stations or towers that can provide information at the ground level were used to study the heat, moisture and momentum in the boundary layer (Stull and Eloranta, 1984; Gentine et al., 2016). In addition, ground aerosol characterization like condensation particle counter (CPC), scanning mobility particle sizer (SMPS), optical particle counter (OPC), and aerodynamic particle sizer

(APS) etc. are used to investigate the aerosol concentration and particle size information (Bates et al., 2000, 2002; Quan et al., 2013; Shin et al., 2014). Mass spectrometry can be used to study the chemical composition of aerosol and gas (Nash et al., 2006; Jordan et al., 2009; Aljawhary et al., 2013). Further more, these *in-situ* instruments can also be deployed on aircraft, balloon, and unmanned aerial vehicles (UAV) to get vertical profiles of aerosol concentrations and components in and above boundary layer (Lenschow, 1986; Greenberg et al., 1999; Neff et al., 2008; Reineman et al., 2016; Kim and Kwon, 2019; Zhang

et al., 2020a).

In addition to these *in-situ* measurements, remote sensing methods including minisodar (Prabha et al., 2002), sonic anemometer (Neff et al., 2008), microwave radiometer (Westwater et al., 1999), as well as lidar (light detecting and ranging) are also used to investigate boundary layers. Lidar is an advanced active sensing instrument that can provide range-resolved and continuous measurements with high temporal (e.g. from seconds to several minutes) and spatial (e.g. from several meters to tens of meters)

resolution. By now, several types of lidar instruments including temperature lidar (Hammann et al., 2015), Doppler wind lidar (Floors et al., 2013), aerosol lidar (Hennemuth and Lammert, 2006), and water vapour lidar (Froidevaux et al., 2013) have been used to measure the boundary layer structures. The temperature lidar can provide the thermal structure of the boundary layer whereas the Doppler wind lidar can offer wind and turbulence information. The aerosol and water vapour lidar can illustrate the distribution of these atmospheric components within the boundary layer. However, most lidars overlap from tens of meters

to around one thousand meters, which makes it difficult to get valid measurements near the surface level for most vertically pointing lidar system. As the height especially of the nocturnal boundary layer varies only from tens of meters to 200 meters (Stull, 1988), it is not easy to determine the structure of the NBL with vertically pointing lidar. But a scanning lidar has the capability to conduct off-zenith measurements or horizontal measurements, hence, allowing to deduce vertical profiles of





aerosols within nocturnal boundary layer.

Recently, synergistic methods combining remote sensing and *in-situ* measurements have been widely used in characterising the PBL and especially the distribution of heat, mass, and momentum within. For example, Kong and Yi (2015) investigated the relationship between the convective boundary layer and surface aerosol concentrations in Wuhan, China, using lidar and OPC. Their findings suggest that the seasonal behaviour of the surface fine particle concentrations mainly depends on the seasonal variation in available volume for aerosol dispersion as given by the convective boundary layer (CBL) height. Cooper

and Eichinger (1994) used lidar and radiosonde data to study the structure of the atmosphere in an urban planetary boundary layer. de Arruda Moreira et al. (2018) compared planetary boundary layer measured by microwave radiometer, elastic lidar and Doppler lidar estimations in the Southern Iberian Peninsula. Lenschow et al. (2012) compared higher-order vertical velocity moments in the convective boundary layer from lidar with *in-situ* measurements and large eddy simulation. Panahifar et al. (2020) monitored atmospheric particulate matter by using lidar, *in-situ* measurements and satellite data over Tehran, Iran.

Large eddy simulation (LES) is a mathematical model for turbulence used in computational fluid dynamics and has been used to simulate atmospheric boundary layers with high spatial resolutions (Mason, 1989; Stoll et al., 2020; Spiga et al., 2021) in the past few years, mainly due to increasing amounts of computational resources being available for research in this field. Khan et al. (2021) developed an atmospheric chemistry model coupled to the turbulence-resolving PALM model system 6.0 (Maronga et al., 2020) (a LES model) to investigate the evolution of gas pollutants ($NO_x$, $O_3$, and CO) in the city of Berlin,

Germany. Slater et al. (2020) investigated the aerosol-radiation-meteorology feedback loop by using a coupled LES in Beijing, which directly qualified the effect of aerosol loading on boundary layer evolution and aerosol mixing process. Wang et al. (2023) investigated air quality in Hongkong combining coupled mesoscale–microscale modeling (WRF-LES-Chem) and *in-situ* sensors to evaluate model performance for different spatial scales. Kurppa et al. (2019) firstly evaluated the vertical variation of aerosol number concentration and size distribution in a simple street canyon without vegetation in Cambridge by

embedding the sectional aerosol module SALSA2.0 (Kokkola et al., 2008, 2018) into the large-eddy simulation model PALM (Maronga et al., 2020). Weger and Heinold (2023) assessed the impact of meteorology and urban topography on the microscale variability of urban air pollution by using LES and empirical orthogonal function (EOF) analysis for the Dresden basin. Their results showed that the model results are strong sensitive to atmospheric conditions, but generally confirm increased eBC levels in Dresden due to the topography. Although the LES has been widely used to study urban boundary layer dynamics, the com-

parison of LES results with observational data especially with high-resolution lidar measurements is rarely done, especially for detailed aerosol particle studies.

The city of Stuttgart is an important industrial centre in southwest Germany with a population of more than 600 000 in a metropolitan area of 2.6 million inhabitants. The city is located in the steep valley of the Neckar river, a basin-like area surrounded by a variety of hills, small mountains, and valleys. The undulating terrain would induce a low wind speed, and weak

synoptic atmospheric circulation which typically hinders the dispersion of aerosol particles (Schwartz et al., 1991; Hebbert et al., 2012). As one of the most polluted cities in Germany, air quality has been a long-standing concern in Stuttgart (Schwartz et al., 1991; Süddeutsche Zeitung, 2016; LUBW, 2016; Huang et al., 2019). The state environmental protection agency, LUBW (Landesanstalt für Umwelt Baden-Württemberg), attributes 58 % of the annual mean $PM_{10}$ at their monitoring station "Am



Neckartor" in downtown Stuttgart to road traffic (45% abrasion, 7% exhaust, 6% secondary formation), 8% to small and
medium-sized combustion sources, and 27 % to the regional background (LUBW, 2019). Mayer (1999) showed the temporal
variability of urban air pollutants (NO, $NO_2$, $O_3$, and $O_x$ (sum of $NO_2$ and $O_3$)) caused by motor traffic in Stuttgart based
on more than 10 years of recorded data, with higher NO concentrations in winter and higher $O_x$ concentrations in summer
(Huang et al., 2019). Kiseleva et al. (2021) investigated nocturnal atmospheric conditions and their impact on air pollutant
concentrations in the city of Stuttgart focusing on the connection between atmospheric conditions and air pollutants using data
from radiosonde, wind lidar, microwave radiometer, and from near-surface meteorological and air quality observations. Samad
and Vogt assessed the effect of traffic density and cold airflows on the urban air quality of a city in Stuttgart with the complex
topography. The results show that the local road traffic emissions account for 52% for $NO_2$ concentrations and 47% for $PM_{10}$
concentration and the city was less polluted when cold airflows blew from west and southwest directions. Figure S1 shows the
seasonal average of $PM_{10}$ in four LUBW monitoring stations and the average values of these four stations in Stuttgart from
2012 to 2022. This figure shows that the concentration of $PM_{10}$ is highest in winter (December, January, and February) and the
monitoring station "Am Neckartor" in downtown Stuttgart shows the highest concentration compared with other monitoring
stations. Hence, this detailed study on the aerosol evolution and its related boundary dynamics near "Am Neckartor" during
winter can improve our understanding of the mechanisms driving air quality dynamics in Stuttgart.

For the research presented in this paper, we collected comprehensive datasets from one field campaign conducted between
February $5^{th}$ and March $5^{th}$, 2018 in downtown Stuttgart and simulation data from LES (PALM-4U) to study the boundary
layer dynamics and air quality in the Stuttgart basin. One scanning aerosol lidar, one wind lidar, one microwave radiometer,
one mobile container equipped with aerosol characterization instrumentation (Huang et al., 2019) and a meteorological sensor,
as well as radiosondes were used in this study. In addition, the large eddy simulation model system, PALM-4U, was used to
simulate the airflow and aerosol evolution in the Stuttgart basin domain over a 48 hour period. The objective of this work is
to study the characteristic evolution of the winter time boundary layer and to investigate the impact of vertical and horizontal
mixing on surface aerosol concentrations by combining the aforementioned datasets. Our study, therefore, adds an important
piece of information on air quality in Stuttgart by investigating the boundary layer dynamics, aerosol chemical composition,
and aerosol physical properties.

This paper is organized as follows. Section 2 describes the remote sensing and *in-situ* methods as well as the implementa-
tion of the PALM-4U model. Details of the evolution of the boundary layer and the impact of mixing processes on aerosol
concentrations are discussed in section 3. In the final section, we provide conclusions.

## 2   Methods

This study is based on a dataset collected in the structured terrain characterizing the city of Stuttgart in southwestern Germany
(c.f. Figure 1). The area of interest includes the relatively broad Neckar valley (width about 2 km), which is orientated from
southeast to northwest, and the basin-shaped valley called the Stuttgart basin (about 2.5 km x 2.5 km), which opens to the
Neckar valley in the northeast. The valley floor is approximately at an altitude of 300 m above mean sea level (m a.s.l.) and





surrounded by hills with ridge heights up to 520 m a.s.l.. A mobile measurement container was installed on a railway bridge in the Rosensteinpark (RSP, 247 m a.s.l., see Figure 1b), downtown Stuttgart. The container was positioned near the opening of the Stuttgart basin to the Neckar Valley, a location characteristic for the urban background. Please note, that the monitoring
station "Am Neckartor" is about 1.5 km southwest from the measurement location used in this study. A scanning aerosol lidar was installed on the roof of this container equipped with *in-situ* instruments including a High-Resolution Time-of-Flight Aerosol Mass Spectrometer (HR-TOF-AMS), an aethalometer (AE 51), a condensation particle counter (CPC), an optical particle counter (OPC, Fidas-200), trace gas sensors and meteorological sensors. For further details on the instrumentation see also Huang et al. (2019). In addition, radiosondes launched at Schnarrenberg (SB, 321 m a.s.l., see Figure 1b) by the Germany
weather service (DWD) provided vertically resolved meteorological parameters. A wind lidar and a microwave radiometer deployed at the Stuttart town hall (TH, 275 m a.s.l., 3.5 km southwest of the measurement container with the lidar, see Figure 1b) measured vertically resolved wind and temperature, respectively. Furthermore, a LES utilizing PLAM-4U (Maronga et al., 2020) was performed to simulate the complex airflow as well as the aerosol distributions in this area.

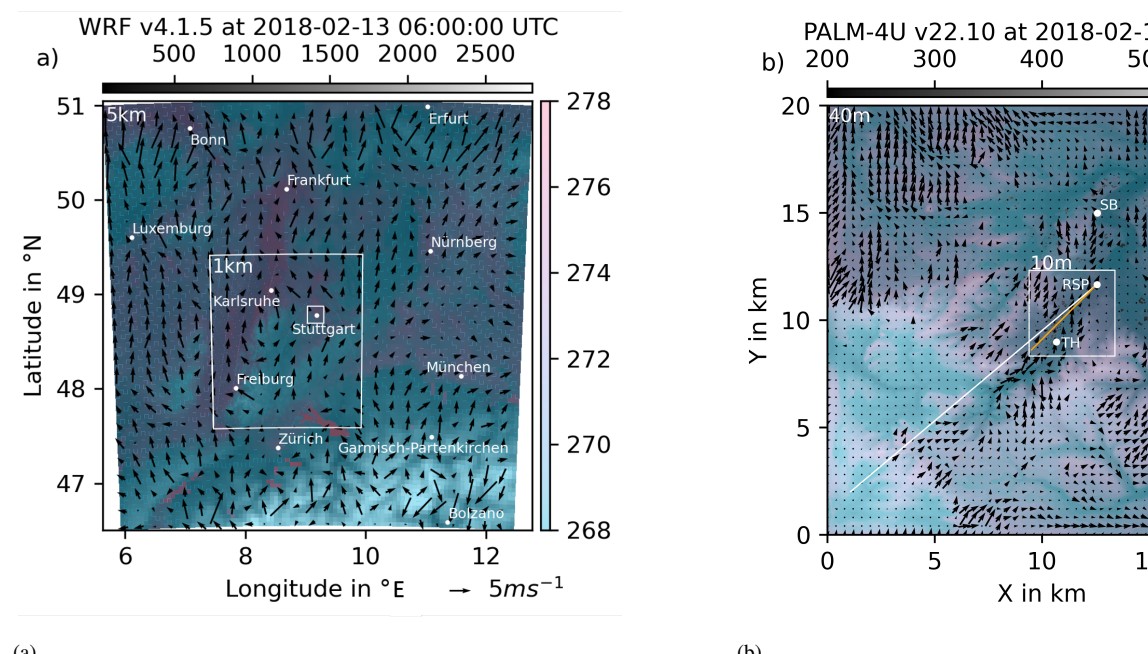

(a)                         (b)

**Figure 1.** Two meter temperatures (contour) and ten meter winds (vectors) from the WRF simulation over the shaded model topography height in m above sea level are shown in (a). The white labels serve for orientation and the white lines mark the approximate domain boundaries. Around Stuttgart the PALM-LES domain boundaries are shown by a small white box. In (b) the PALM-4U domains are presented using the same type of visualization for the same model output time. Shown are potential temperature and horizontal winds on the second model level above surface, (i.e., 15 meter a.g.l.). The labels indicate measurement site locations and the white line indicates the aerosol laser scan beam, while the orange line indicates the location of the vertical section evaluated from PALM-4U (RSP = Rosenstein Park).



## 2.1 Remote sensing

### 2.1.1 Scanning aerosol lidar

The scanning aerosol lidar (Rametrics Inc., Type: LR111-ESS-D200, named KASCAL) used in this campaign is a mobile scanning system with an emission wavelength of 355 nm. The laser pulse energy and repetition frequency are 32.1 mJ and 20 Hz, respectively. The laser head, 200 mm telescope, and lidar signal detection units are mounted on a rotating platform allowing zenith angles from -7 º to 90 º and azimuth angles from 0 º to 360º. This lidar works automatically, scheduled,

and continuously via software developed by Raymetrics. Detailed information can be found at https://www.raymetrics.com/product/3d-scanning-lidar, last access: 1 September 2022 (Avdikos, 2015; Zhang et al., 2022). The lidar was put on the roof of the container and conducted zenith scanning measurement with elevation angle from $90°$ to $5°$ with the step of $5°$. The beam of the lidar was directed along the basin axes as shown as a white line in Figure 1.

For the data analysis and calibration of the system, we followed the quality standards of the European Aerosol Research Lidar

Network (EARLINET) (Freudenthaler, 2016). The data analysis for the zenith measurements employed the Klett-Fernald method to obtain vertical profiles of backscatter coefficients. These vertical aerosol backscatter coefficients were used as the reference values for other elevation angle measurements.

The atmospheric boundary layer heights were determined from lidar data by using the haar wavelet transform (HWT) method (Pal et al., 2010). The method is defined as

$$z_{HWT} = \max[w_f(a,b)] = \max \frac{1}{a} \int_{z_{min}}^{z_{max}} X(z)H(\frac{z-b}{a})\,dz \tag{1}$$

In which $W_f$ is the covariance transform value, $X(z)$ is range corrected lidar signal defined as $X(z) = P(z) * z^2$, and $H(\frac{z-b}{a})$ is the Harr Wavelet function as defined as followed:

$$H(\frac{z-b}{a}) = \begin{cases} 1 & b - \frac{a}{2} \leq z \leq b \\ -1 & b \leq z \leq b + \frac{a}{2} \\ 0 & \text{elsewhere} \end{cases} \tag{2}$$

The dilation a is set to be 75 m in this paper. $Z_{min}$ and $z_{max}$ are the lower and upper heights for the lidar signal profile,

respectively.

### 2.1.2 Wind lidar

The wind lidar principle relies on the measurement of the Doppler shift of laser radiation backscattered by the particles in the air (dust, aerosols). The WindCube v2 (Leosphere - Vaisala) measures wind speed with a Doppler beam swinging (DBS)

technique (Rao et al., 2008), where an optical switch is used to point the lidar beam in the four cardinal directions (north, east, south, and west) at an elevation angle of $62°$ from the ground and it allows us to get 200 m vertical wind profiles of wind speed



and direction, turbulence, and wind shear. Detailed information about wind cube is given on the Vaisala homepage (Vaisal, 2021).

### 2.1.3 Microwave radiometer

The microwave profiler HATPRO was manufactured by Radiometer Physics GmbH, Germany (RPG) as a network-suitable microwave radiometer with very accurate retrievals of Liquid Water Path (LWP) and Integrated Water Vapor(IWV) at high temporal resolution (1 s). The spectral characteristics of the instrument also make it possible to observe the temperature profile and to a limited extent also the humidity profile (Löhnert and Maier, 2012).

### 2.2 In situ measurements

The meteorological sensors, WS700 (Lufft GmbH), provided air temperature, relative humidity, wind direction, wind speed, global radiation, pressure, and precipitation data. Different trace gases ($O_3$, $CO_2$, $NO_2$, $SO_2$) sensors measured corresponding gas compositions. An Aethalometer (AE51; Aethlabs Inc.) measured temporal variability of equivalent black carbon (eBC) concentrations (Petzold et al., 2013). An HR-ToF-AMS equipped with an aerodynamic lens (Williams et al., 2014) was installed in a mobile container to continuously measure total non-refractory particle mass as a function of size (up to 2.5 $\mu$m

particle aerodynamic diameter) at a temporal resolution of 30 s. The AMS inlet was connected to a $PM_{2.5}$ head (flow rate 1 m$^3$ h$^{-1}$; Comde-Derenda GmbH) and a stainless-steel tube of 3.45 m length. The AMS data were analyzed with AMS data analysis software packages SQUIRREL (version 1.60C) and PIKA (version 1.20C). Positive matrix factorization (PMF; (Paatero and Tapper, 1994; Paatero, 1997)) was applied for AMS data to identify different aerosol source factors for source appointment (Ulbrich et al., 2009; DeCarlo et al., 2010; Zhang et al., 2011; Mohr et al., 2012; Canonaco et al., 2013; Crippa et al.,

2014; Shen et al., 2019). This allows to differentiate organic aerosol into e.g. hydrocarbon-like OA (HOA), cooking-related OA (COA), nitrogen-enriched OA (NOA), biomass burning OA (BBOA), semi-volatile oxygenated OA (SV-OOA), and low-volatility oxygenated OA (LV-OOA). The mass spectra of these five OA factors resolved from the PMF analysis are shown in Figure S2. These *in-situ* data were averaged over 10 minutes. Detailed information about *in-situ* measurements and aerosol chemical composition are introduced in Huang et al. (2019).

In addition to *in-situ* container measurements, radiosondes at Schnarrenberg meteorological station (SB, see Figure 1b) were launched by the German Weather Service (DWD) to measure the vertical profile of meteorological parameters (e.g. Temperature, humidity, pressure, wind). The vertical profiles of temperature and humidity were used to determine boundary layer heights. Detailed descriptions of boundary layer retrieved methods were introduced in previous publications (Hennemuth and Lammert-Stockschlaeder, 2006; Liu and Liang, 2010; Seidel et al., 2010; Guo et al., 2016).






## 2.3 Large eddy simulation

PALM-4U (Maronga et al., 2020) is a model system that has been developed to simulate a wide range of urban micro-scale processes. The center of this model system is the large eddy simulation model PALM (Raasch and Schröter, 2001) based on non-hydrostatic, filtered, incompressible Navier-Stokes equations in Boussinesq-approximated form. To force this microscale model for realistic cases, meteorological data is required for initial- and boundary conditions, as well as detailed information about the modeled surface properties (e.g. topography). Details of the model pipeline are described below.

## 2.4 WRF Setup

The Weather Research and Forecasting (WRF) model Version 4.1.3 (Skamarock et al., 2021) was forced by ERA5 reanalysis data (Hersbach et al., 2020) and Local Climate Zone (LCZ) data (Demuzere et al., 2022a, b) to produce consistent meteorological fields from 11 February to 14 February, 2018 as forcing for the microscale simulation. Two nested domains with 5 km and 1 km horizontal grid-spacing have been placed, such that Stuttgart is located at the center and a sufficiently large part of the European continent is covered, to allow for all relevant flow fields to evolve appropriately.

ERA5 is the fifth generation European Centre for Medium-Range Weather Forecasts (ECMWF) atmospheric reanalyses of the global climate (Hersbach et al., 2020). ERA5 provides multiple climate variables at a spatial resolution of 0.25 degrees (approximately 30 km) for the globe every hour, with 137 levels from the surface up to 0.01 hPa (around 80 km height) (https://doi.org/10.24381/cds.adbb2d47, last access: 23 September 2023).

## 2.5 PALM-4U Setup

For the successful simulation of the complex, topographically forced flows around Stuttgart, a relatively large model domain is required. Two nested domains spanning 20 by 20 km and 4 by 4 km, with 40 m and 10 m grid-spacing respectively have been set up. The static data required for these two domains was described by Heldens et al. (2020). The output of the WRF simulation was processed with the PALM-4U package tools for the 48 hour period from 12 to 14 February, 2018 to create initial- and boundary conditions to force PALM-4U. Wind, temperature, moisture, radiative fluxes and soil variables were assimilated from this WRF data.

Particulates ($PM_{10}$) were simulated with the phstatp chemical mechanism, which allows for emissions, transport and dry deposition (Kurppa et al., 2019). The emissions were parameterized by street types and initial profiles approximated from observed profile values at initialization time. These profiles persisted as constant boundary conditions for the entire 48 hour period. Note, that the nested domain is located at a distance of approximately 8 km from the outer boundary, at which this constant nocturnal profile is forced. During stable nocturnal conditions, the profile properties are mostly conserved throughout the transport process (assuming small vertical transport). Convective daily conditions produce adequately mixed particulate profiles at the child domain's boundary, due to the sufficient distance (larger than three times the boundary layer height). This simplified approach leads to particulate concentration fields, which approach a balance between dry deposition and emission.





The model output was averaged in time over 10-minute intervals and put out above the model surface (i.e., terrain following) up to heights of 1500 m a.g.l. to maximize compatibility with the measured data.

## 3   Results and Discussion

In this section, we will firstly review the measurements during this field campaign. Then we will discuss our result on the correlation of boundary layer heights with ground level aerosol concentrations, especially the relationship at nighttime. Afterwards, two selected cases were used to demonstrate the boundary layer evolution and aerosol mixing processes within the boundary layer. Finally, the LES (PALM-4U) was used to simulate the boundary layer processes and to investigate the aerosol mixing processes within the boundary layer in the context of the local and regional flow properties.

Figure 2a shows time series of the range corrected lidar signal (RCS) for the whole observation period as well as boundary layer heights retrieved from lidar during periods that are cloud free up to 3 km a.g.l.. In addition, boundary layer heights derived from radiosonde and ERA5 are also shown in this figure as indicated by stars and black dashed line respectively. This panel shows a good agreement in boundary layer heights among lidar and radiosonde measurements as well as the ERA5 dataset. The correlation of boundary layer height between lidar and radiosonde measurements is shown in the left panel of Figure S3,

which shows that the boundary layer heights retrieved from lidar and radiosonde agree well with each other with a slope of $1.10 \pm 0.14$ and a Pearson correlation coefficient of 0.86. The correlation of boundary layer heights between lidar and ERA5 reanalysis is shown in the right panel of Figure S3, which shows a slope of $0.70 \pm 0.07$ and a Pearson correlation coefficient of 0.61. The boundary layer heights from the ERA5 reanalysis are systematically lower than that from lidar and radionsonde retrieval but still show the same trend as the lidar measurements. This underestimation was also reported by Dias-Júnior et al.

(2022). The evolution of aerosol composition measured by HR-TOF-AMS and eBC concentrations are shown in Figure 2b. The data indicates that nitrates are dominant in aerosol chemical composition due to high NOx emissions and lower air temperatures in winter inhibiting evaporation of ammonium nitrate (Xie et al., 2020; Zhang et al., 2020b). Positive matrix factorization (PMF) analysis of organic aerosol (OA) factors shown in Figure 2c illustrates that low-volatility oxygenated organic components (LV-OOA) are dominant during these measurements. These compounds are mostly attributed to aerosol from regional

transport (Song et al., 2022). A detailed analysis of the chemical composition of the aerosol in Stuttgart can be found in Huang et al. (2019). The average temperatures at two altitude ranges (0.5-1.0 km and 1.0-1.5 km) measured by radiosonde and wind speed at 10 m above ground level measured by the meteorological sensor (WS700) is shown in 2e. The temperature inversion (red area between two temperature lines) and low wind speed periods coincides with an accumulation of aerosols (e.g. from February $6^{th}$ to February $8^{th}$ and from February $28^{th}$ to March $2^{th}$). The obvious temperature inversion and low wind speeds

during the above two periods are labled as stagnant meteorological conditions, which suppressed convection in the troposphere, hence causing a shallow and nocturnal boundary layer and accumulation of aerosols at ground level. Stagnant conditions are also an important reason of air pollution in mega cities (Huang et al., 2018; Katsoulis, 1988; Ji et al., 2014).

Figure S4 shows the vertical profile of temperature (left) and wind speed (right) during the polluted period and a less polluted period. The polluted period is defined for concentration of $PM_{10}$ exceeding the ambient air quality standard for the Euro-







**Figure 2.** Time series of range corrected lidar signal (contour) and boundary layer heights derived from scanning aerosol lidar (pink line), radiosonde (stars), and ERA5 dataset (black dashed line) (a), the aerosol mass concentrations for different chemical components (b), five-factor PMF solutions of organic aerosol (c), the particle matter concentrations measured by OPC (d), and the temperatures at two different altitude levels measured by radiosonde as well as wind speed measured at 2 m above the ground level (e).

pean Union (25 $\mu$g/m$^3$; https://www.transportpolicy.net/standard/eu-air-quality-standards/, last access: 3 July 2023), which is indicated as the grey dashed line in Figure 2d. These average profiles of temperature and wind speed shown in Figure S4 were calculated after excluding the data collected on weekends to avoid the influence of local emission differences between weekdays and weekends. Figure S4 also shows the stagnant dynamics effects on air pollution as described above.





### 3.1 Correlation between boundary layer heights and ground-level aerosol concentrations

Figure S5 (a, e, i) shows the correlation between boundary layer heights and $PM_{10}$, eBC, as well as BBOA concentrations for three different subsets of data, respectively. The color of the scatter points indicates the relative humidity. For all $PM_{10}$ data points an anti-correlation as shown in figure S5a, was found for boundary layer heights above 900 m. This anti-correlation means that a deeper boundary layer diluted the aerosol while a shallower boundary layer concentrated aerosol at the ground level. However, we also found a positive correlation between $PM_{10}$ and boundary layer heights for boundary layer heights below 900 m (a.s.l.). This positive correlation is also reported in Yuval et al. (2020) and typically coincided with low wind speed and high relative humidity, indicating typical properties of the nocturnal boundary layers.

Then the data was divided into three groups for three different time periods - morning (04:00 - 10:00 UTC) (b, f, j), afternoon (12:00-18:00 UTC) (c, g, k), and night (18:00 - 04:00 UTC) (d, h, l). The correlation between the boundary layer and surface aerosol concentrations ($PM_{10}$) in the these three subplots (b, c, d) show a positive correlation for PBL heights below 900 m (a.s.l.) and a weaker but negative correlation for larger PBL heights. The correlation between boundary layer heights and eBC as well as BBOA concentrations shown in Figure S5 revealed that the eBC and BBOA concentrations are always anti-correlated with the boundary layer heights. The reason for the positive correlation between $PM_{10}$ and boundary layer height below 900 m a.s.l. is due to the local emissions and aerosol water take up during night and early morning. The reason for only anti-correlation between the boundary layer heights and eBC as well as BBOA concentrations is that the eBC and BBOA particles emission from sources like biomass burning or traffic are smaller and less hygroscopic and thus could be diluted by boundary layer evolution.

A good case to illustrate this phenomenon is shown in Figure 4. The chemical composition measured by the AMS is shown in Figure 4b. From this figure, we found that the mass concentration of various aerosol components (e.g. ammonium sulfate, ammonium nitrate) increased from February $13^{th}$, 18:00 to February $14^{th}$, 5:00 while the boundary layer heights increased slowly during this time period, which caused a positive correlation between the boundary layer and $PM_{10}$ concentration. However, the eBC and BBOA concentrations shown in Figure 4c and Figure 4d are constant in the nighttime. Hence the $PM_{10}$ concentrations can be correlated with boundary layer heights while eBC and BBOA concentrations are always anti-correlated with boundary layer heights.

The above statistical data analysis of the correlation between ground-level aerosol concentrations and the boundary layer heights is based on data collected during one month. More data were analysed to support this relationship. Figure S6 shows the diurnal variations of $PM_{10}$ and the boundary layer heights based on two-year data from January $1^{st}$, 2020 to December $31^{st}$, 2021 in Stuttgart. The $PM_{10}$ concentrations are the hourly reported dataset by LUBW and the boundary layer heights are from an ERA5 dataset. Figure S6 shows a positive correlation between boundary layer heights and $PM_{10}$ concentrations between 04:00 - 08:00, UTC as shaded in Figure S6, and this positive correlation is possibly related to local morning emission or water take up during morning rush hours. In addition, the increasing boundary layer after sunrise (08:00 - 12:00) diluted the aerosol within the boundary layer, thus causing a decrease of $PM_{10}$ concentrations.

The diurnal variations of $PM_{10}$ concentrations and the boundary layer heights are shown in Figure 3 for the winter of 2018





**Figure 3.** Diurnal variations of PM$_{10}$ concentrations (black) and the boundary layer heights (blue) for the winter of 2018 based on our measurement (top panel) as well as for different seasons (Winter: DJF, Spring: MAM, Summer: JJA, Spring: SON) based on two years data from January $1^{st}$, 2020 to January $1^{st}$, 2022 in Stuttgart. The two-year PM$_{10}$ concentrations are hourly reported data by LUBW and the boundary layer heights are from ERA5 data. The grey shaded time interval shows correlations between BLH and PM10 for all seasons.

based on our measurement (top panel) as well as different seasons based on LUBW- and ERA5-data. This also shows that the ground-level PM$_{10}$ concentrations are correlated with boundary layer heights from 04:00 to 08:00 for all datasets. However, the strength of the correlation is different for different seasons. The spring (MAM) shows the strongest correlation (Pearson correlation coefficient: 0.83) while the winter (DJF) shows the weakest correlation (Pearson correlation coefficient: 0.26). In addition, the summer has the highest mixing layer height (1283 ± 399 m) while the winter has the lowest mixing layer height (682 ± 542 m) as expected due to the solar radiation being strongest in summer while weakest during winter. The ground-level PM$_{10}$ aerosol concentrations are anti-correlated with that mixing layer heights and shows the highest concentrations during



winter ($33 \pm 32$ $\mu$g/m$^3$) and the lowest concentrations during summer ($16 \pm 7$ $\mu$g/m$^3$). From the correlation between PM$_{10}$ concentrations and boundary layer heights, we conclud that the ground-level PM$_{10}$ concentrations are anti-correlated with mixing layer heights but correlated with nocturnal boundary layer heights.

### 3.2  Boundary layer dynamics and surface level aerosol - case studies

In Figure 2, the evolution of the boundary layer heights and their effect on surface aerosol mixing processes is illustrated for
the whole measurement period. In this section, two cases are selected to demonstrate these processes in detail. Figure 4a shows the time series of lidar retrieved vertical backscatter coefficients, the boundary layer heights (white solid line), the residue layer heights (white dashed line), and the boundary layer heights from the ERA5 dataset (grey dashed line) as well as the boundary layer heights retrieved from radiosonde (yellow triangles). Please note that the altitude used here is the height above sea level. The reason for using altitude instead of height above ground level is that the altitudes of these three observation stations are
different as shown in Figure 1a. The vertically distributed backscatter coefficients are shown from ground level to the free troposphere by merging zenith and near horizontal (5° above the horizon) measurements. The time series of aerosol chemical composition measured by AMS is shown in Figure 4b and the PMF analysis result of the organic aerosol with 5 factors is shown in Figure 4c. In addition, the potential temperatures ($\theta$) from the microwave radiometer and turbulent kinetic energy (TKE) from wind lidar data are shown in Figure 4e and Figure 4f, respectively. Finally, the eBC concentrations and solar radiation are
shown in Figure 4d and Figure 4g, respectively.

The vertically extended backscatter coefficients in this figure show that most aerosol can only reach a maximum height of 1800 m (below boundary layer height) for the whole period, which indicates that most of the aerosol only stayed within the boundary layer or residual layer and could not reach to the free troposphere as stated in previous literature (Guo et al., 2009; Quan et al., 2013; Li et al., 2017; Su et al., 2018; Yuval et al., 2020). We also found that the mixing layer heights measured by lidar and
radiosondes show a good agreement in this case.

A decreasing trend of the residual layer (RL) height and a weakly increasing PBL height at around $550 \pm 93$ m (a.s.l.) can be seen during night time. The shallow and nocturnal boundary layer and increased emissions during morning rush hours (5:00 a.m. - 10:00 a.m.) caused a rapid accumulation of aerosol near the surface as can be seen from low-attitude backscatter coefficients and ground level *in-situ* measurements. Driven by increased solar radiation after 10:00 on February 14$^{th}$ the boundary
layer height increased and diluted the aerosol within the boundary layer, thus causing a decrease of aerosol concentrations at ground level. Furthermore, we found that the aerosol concentrations increased more during morning rush hours (5:00 - 10:00) than during evening rush hours (17:00 - 20:00) mainly due to the shallow boundary layer in the morning. The increased aerosol during morning and evening rush hours is related to the emissions of traffic (HOA) and industry (Amine-OOA) as can be seen from the PMF analysis results shown in panel (c). At night time, the potential temperature inversion shown in panel (e) and a
small value of turbulent kinetic energy (TKE) shown in panel (f) indicate a stable and shallow boundary layer.

Figure S7 show the similar plot as Figure 4 but with the ground level aerosol concentration normalized by the boundary layer heights (Huang et al., 2023; Tsai et al., 2011). This figure shows that the total non-refractory particle mass (especially nitrates) increased from 3.9 $\mu$g/m$^3$ to 10.8 $\mu$g/m$^3$ morning rush hours on February 14$^{th}$, 2018. While during the night time (18:00 -





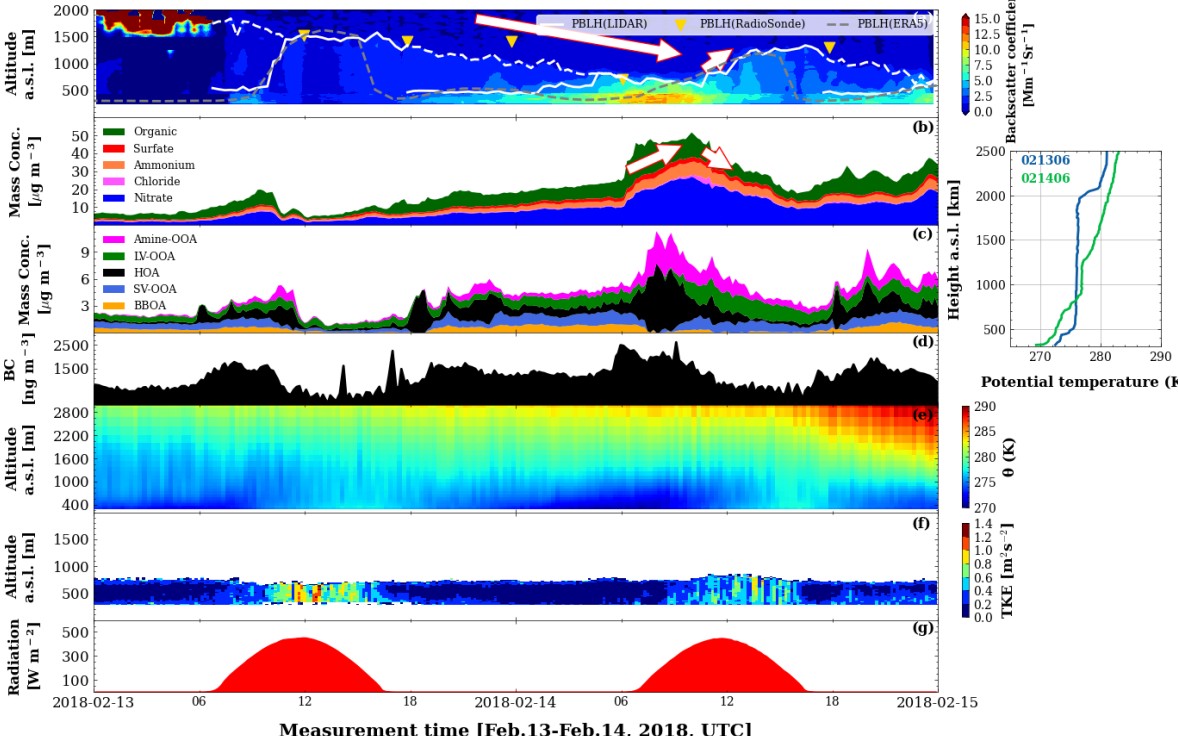

**Figure 4.** Time series of backscatter coefficients from lidar measurements (contour plot), the boundary layer heights from lidar measurement (white solid line), the ERA5 dataset (grey dashed line), and DWD radiosonde (yellow triangle) as well as residual layer heights retrieved from lidar (white dashed line) (a), the aerosol mass concentrations measured by aerosol mass spectrometer (AMS) (b), five-factor positive matrix factorization (PMF) solutions of organic aerosol(c), black carbon concentrations (d), potential temperature measured by microwave radiometer (MWR) (e), turbulence kinetic energy (TKE) retrieved from Doppler lidar (f) as well as the global radiation measured by meteorological sensors (WS700) (g) for case 1 from February $13^{th}$ to February $14^{th}$, 2018. The white arrows in panel a and b show the decreasing or increasing trends of boundary layer height. The plot on the right side shows the potential temperatures measured by radiosonde at 06:00 of $13^{th}$ and $14^{th}$, February, 2018.

.

04:00, UTC), the nitrate aerosol particles increase from 2.5 $\mu$g/m$^3$ to 3.9 $\mu$g/m$^3$ while the eBC concentrations showed a slight
decrease from 1048 ng/m$^3$ to 464 ng/m$^3$. The aerosol horizontal transport source was not considered as the wind speed is 0.76
$\pm$ 0.35 (less than 1 m/s in most of time) from February $13^{th}$, 2018, 18:00 to February $14^{th}$, 2018, 12:00. The only considered
source during this period is local emission. The reason for the increase of non-refractory particles concentrations but the decrease of eBC concentration is that the non-refractory particles were emitted during the night time or take up water due to high
relative humidity while the emissions of eBC particles was diluted due to slight increase of nocturnal boundary layer during
the night time (Su et al., 2020).



Interestingly, we found in this case, that the boundary layer height increased slower on the second day (February $14^{th}$) than that on the first day (February $13^{th}$). There was a time delay in the boundary layer convection during the second day despite the solar radiation being the same on these two days (Fig 4g). The reason for these different boundary layer evolutions is due to different vertical thermal structures as can be seen in the vertical temperature profiles given in Figure 4d and Figure 4 insert.

On the first day, the temperature inversion is weaker and the TKE is larger than on the second day as can be seen from Figure 4e and Figure 4f, which means that it takes a shorter time to transform the nocturnal boundary layer into the convective boundary layer. Hence, the boundary layer increases faster on the first day than on the second day. One explanation of these thermal structure differences for these two days is the presence of clouds during the first night. They prevented longwave emissions and weakened the temperature inversion, which caused a neutral boundary layer during night time. Furthermore, the boundary

layer increased faster after sunrise due to this neutral boundary layer in the morning. Finally, the delay of the boundary layer convection process on the second day prevented diffusion of aerosol during morning rush hour (5:00 - 10:00, February $14^{th}$), thus causing accumulation of aerosol at ground level as shown in Figure 4a-c. The conceptual schematic for this phenomenon is summarized in Figure 5.

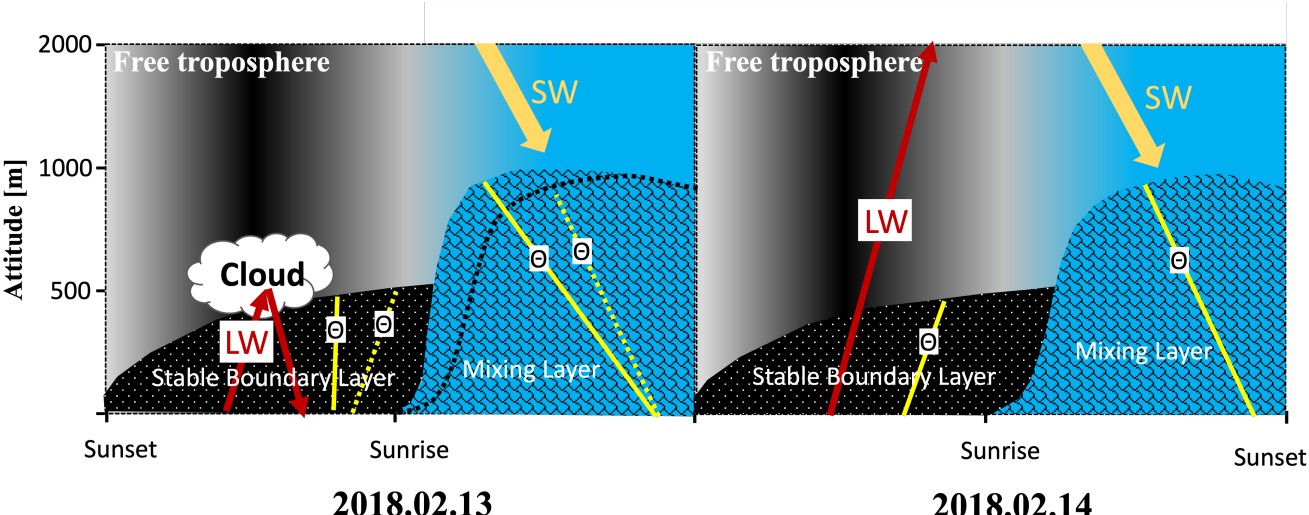

**Figure 5.** Concept of boundary layer and the role of clouds on boundary layer evolution. The gold arrows indicate short-wavelength radiation; the red arrows indicate the long-wavelength radiation; the yellow solid lines indicate the potential temperature; the yellow dotted lines on the left side indicate the potential temperature on February $14^{th}$ for comparison; the black textured areas indicated the stable boundary layer; the blue textured areas indicate mixing layer; the black dotted line on the left side indicates the boundary layer height on February $14^{th}$ for comparsion; LW: Long Wavelength radiation, SW: Short Wavelength radiation, $\theta$: Potential temperature.

.

Figure 6 shows the results during case 2, in which the same methods as in case 1 were applied, but showed different patterns.

The most obvious phenomenon is that a sharp decrease in aerosol concentrations from 7:00 to 12:00, February $24^{th}$, was ob-




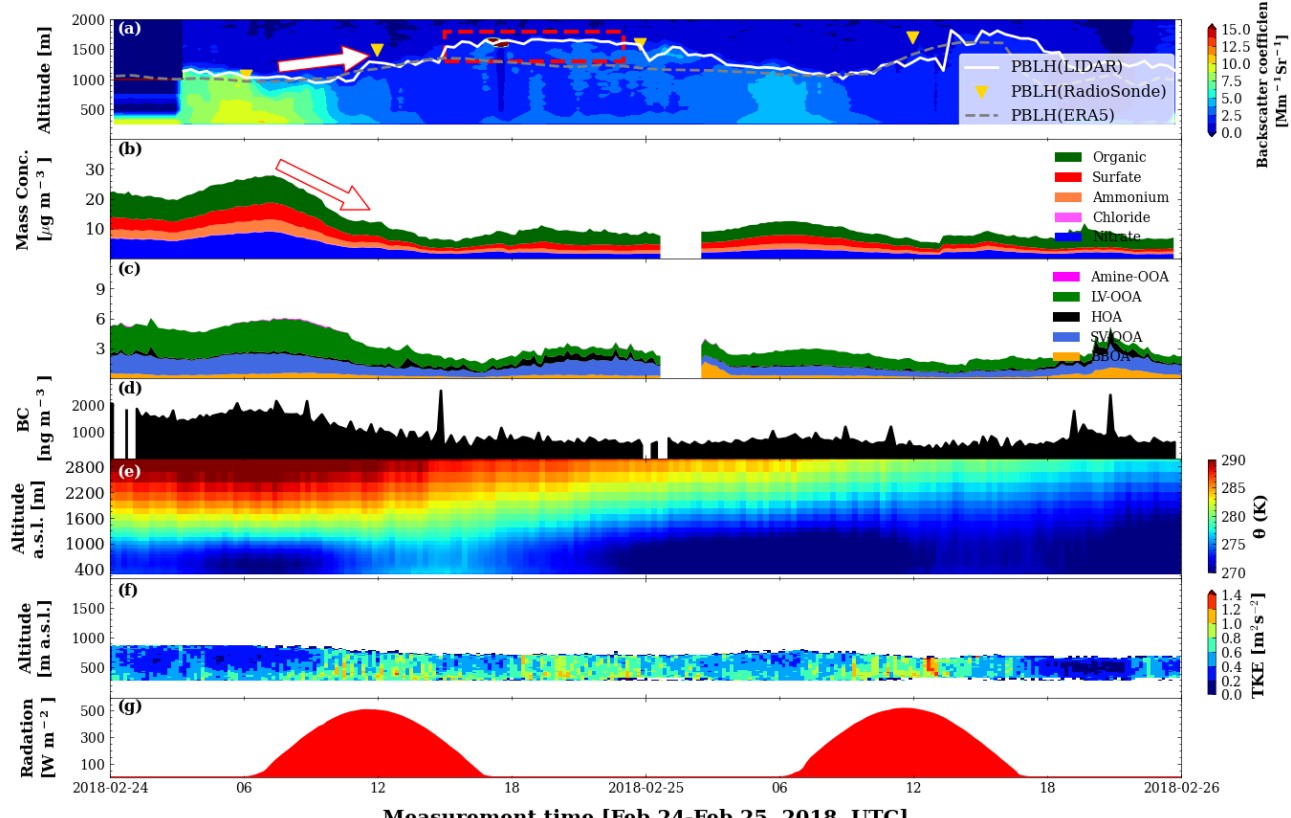

**Figure 6.** Time series of backscatter coefficients from lidar measurements (contour plot), the boundary layer heights from lidar measurement (white solid line), the ERA5 dataset (grey dashed line), and DWD radiosonde (yellow triangle) (a), the aerosol mass concentrations measured by aerosol mass spectrometer (AMS) (b), five-factor positive matrix factorization (PMF) solutions of organic aerosol (c), black carbon concentrations (d), potential temperatures measured by microwave radiometer (MWR) (e), turbulence kinetic energy (TKE) retrieved from Doppler lidar (f) as well as the global radiation measured by meteorological sensors (WS700) (g) for case 2 from February $24^{th}$ to February $25^{th}$, 2018.

served even though the boundary layer heights only increased from 1042 m a.s.l. to 1280 m a.s.l. In addition, the boundary layer heights did not decrease after sunset of February $24^{th}$ as shown in the red rectangle. Furthermore, the boundary layer heights measured by radiosonde is higher than those derived from lidar measurement, in contrast to case 1 (Figure 4). We also found that the aerosol concentrations at ground level were much lower on February $25^{th}$ than those on February $24^{th}$, corresponding to a higher boundary layer on February $25^{th}$. Finally, the PMF analysis result shown in panel (c) shows a large fraction of LV-OOA for organic composition, which is typically more related to regional transport (Song et al., 2022).

Figure 4 and Figure 6 showed different boundary layer evolutions and different patterns of aerosol concentrations at ground level even with a similar evolution of solar radiation as shown in panel (g) of both figures. However, the evolution of temper-






ature and wind are different as shown in Figure S8. Comparing the meteorological background in these two cases, we found

that the temperature decreased more rapidly for case 2 as shown in the bottom panel of this figure. This decrease caused a

lower temperature on February $25^{th}$. The temperature was below 0°C even during daytime of February $25^{th}$. In addition, a

higher wind speed was observed from 07:00, February $24^{th}$ to 16:00, February $25^{th}$ for case 2. Then, the wind speed began

to decrease and the wind direction also changed from east to north since 16:00, February $25^{th}$. All these meteorological in-

formation indicate that a cold front passed by the observation station from February $24^{th}$ and February $25^{th}$ affecting local

temperature and wind, thus having an impact on the boundary layer evolution and aerosol distributions in the boundary layer.

The high wind speed during this cold front causes strong turbulence in the boundary layer, thus increasing the boundary layer

heights, especially at nighttime. In addition, this high wind speed also blew the local aerosol away and caused a low aerosol

concentration on February $25^{th}$.

From the above two cases, we conclude that the evolution of the boundary layer was affected by related meteorological factors

such as solar radiation, clouds, wind speed and wind direction, which in turn affects the aerosol distribution in the boundary

layer.

### 3.3    Comparison of large eddy simulations with observations

Case 1 outlined above is a good example of boundary layer evolution and aerosol mixing processes for two consecutive days.

The comprehensive dataset collected during this case provided us a good opportunity to validate the LES model PLAM-4U.

As the coordinate used in the model is the height above the ground level. To be simplified, we use the altitude above the ground

level (a.g.l.) to compare observation result with model simulation.

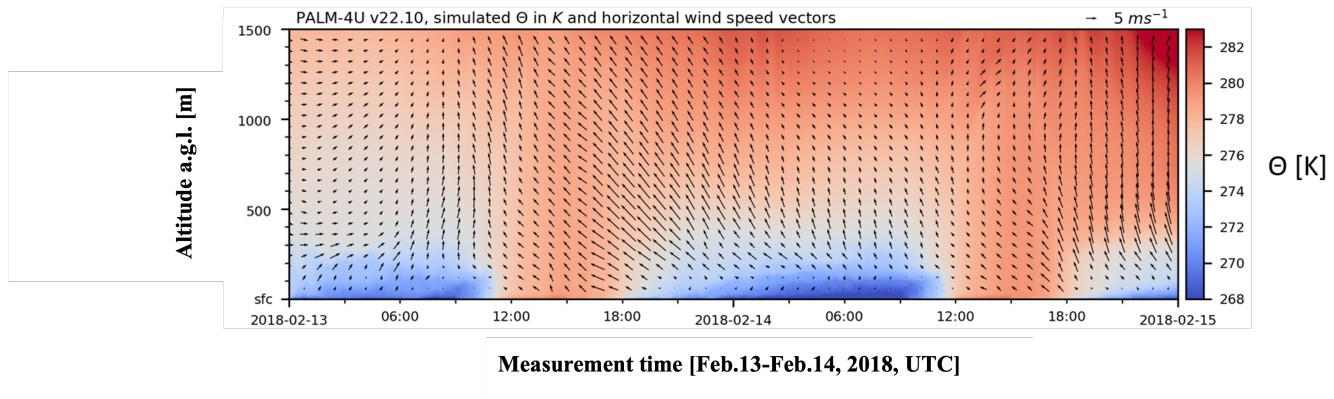

**Figure 7.** Time-height-section of simulated potential temperature in K and horizontal wind in m s$^{-1}$ at the RSP site from February $13^{th}$ to
February $14^{th}$, 2018 from the surface to 1500 m a.g.l..

In Figure 1, both the mesoscale and micro scale model results are shown side-by-side to illustrate the added benefit of sim-

ulating microscale processes for representing local effects. Note the warming effect in the Rhine valley simulated by WRF,

and the cold pooling effects in the Neckar valley and other smaller valleys simulated by PALM-4U. The diurnal development



of the boundary layer temperature fields as simulated by PALM-4U is shown in Figure 7 as the time-height section above the RSP site as indicated in Figure 1b. Nocturnal cooling near the surface underneath the residual heat from the daytime and the stabilization of the boundary layer was captured by the model dynamics. During daytime, neutral, convective conditions were simulated with reoccurring stabilization, after long-wave radiative cooling outweighed short-wave radiative heating at the surface. We found these thermally driven circulation processes to be simulated in a plausible way qualitatively, based on

the observational data we compared the simulation results to. An exact quantitative comparison and attribution of deviations has not been part of this study, as the focus was on the measured data. Such an analysis will follow once more cases can be compared.

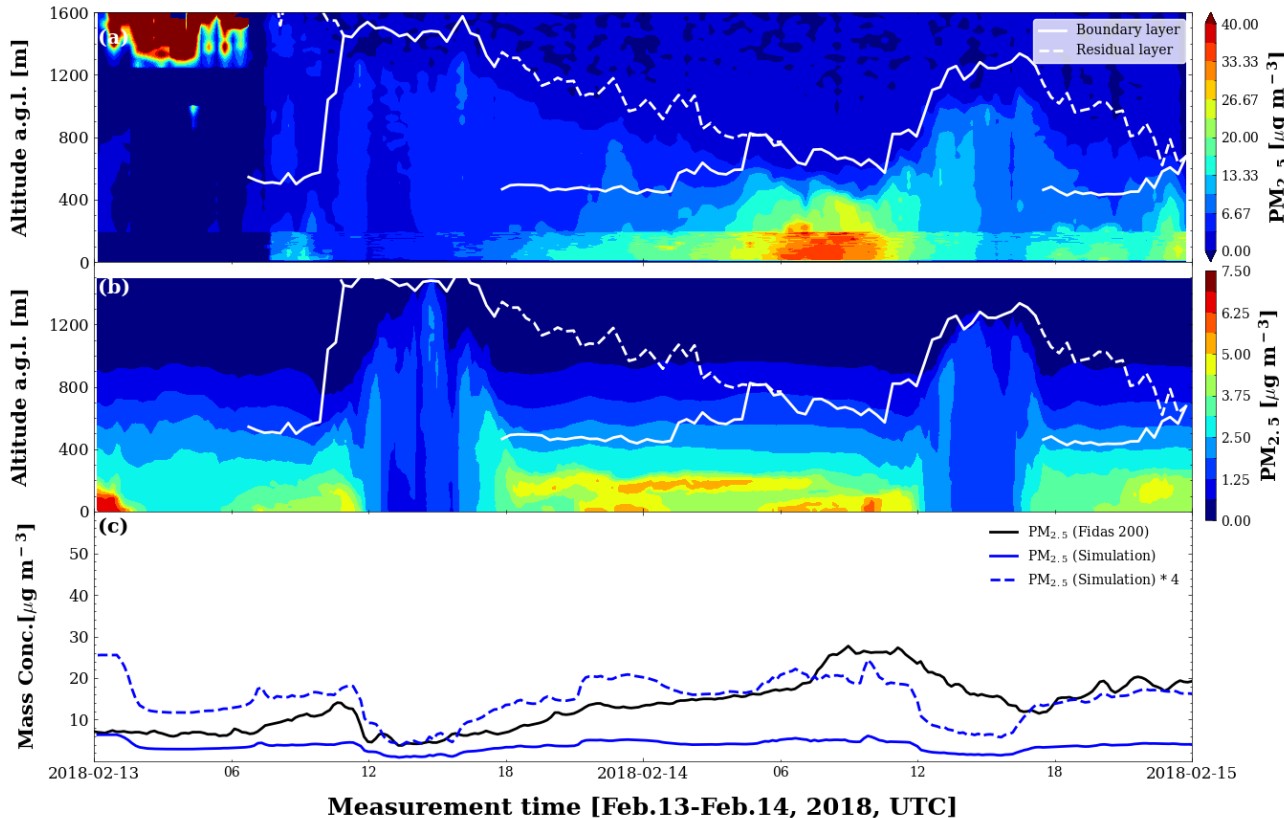

**Figure 8.** Time series of the PM$_{2.5}$ concentrations retrieved from lidar measurements (a) and PALM-4U (b), the boundary layer height (white line) and residual layer height (white dashed line) retrieved from lidar, as well as ground level PM$_{2.5}$ concentration measured by Fidas200 and modeled by PALM-4U (c) for case 1 from February 13$^{th}$ to February 14$^{th}$, 2018.

Figure 8 shows the time series of the PM$_{2.5}$ concentrations retrieved from lidar measurements and PALM-4U. In this case,

the PM$_{2.5}$ concentrations from lidar measurement are converted from lidar-derived extinction coefficients by using the con-



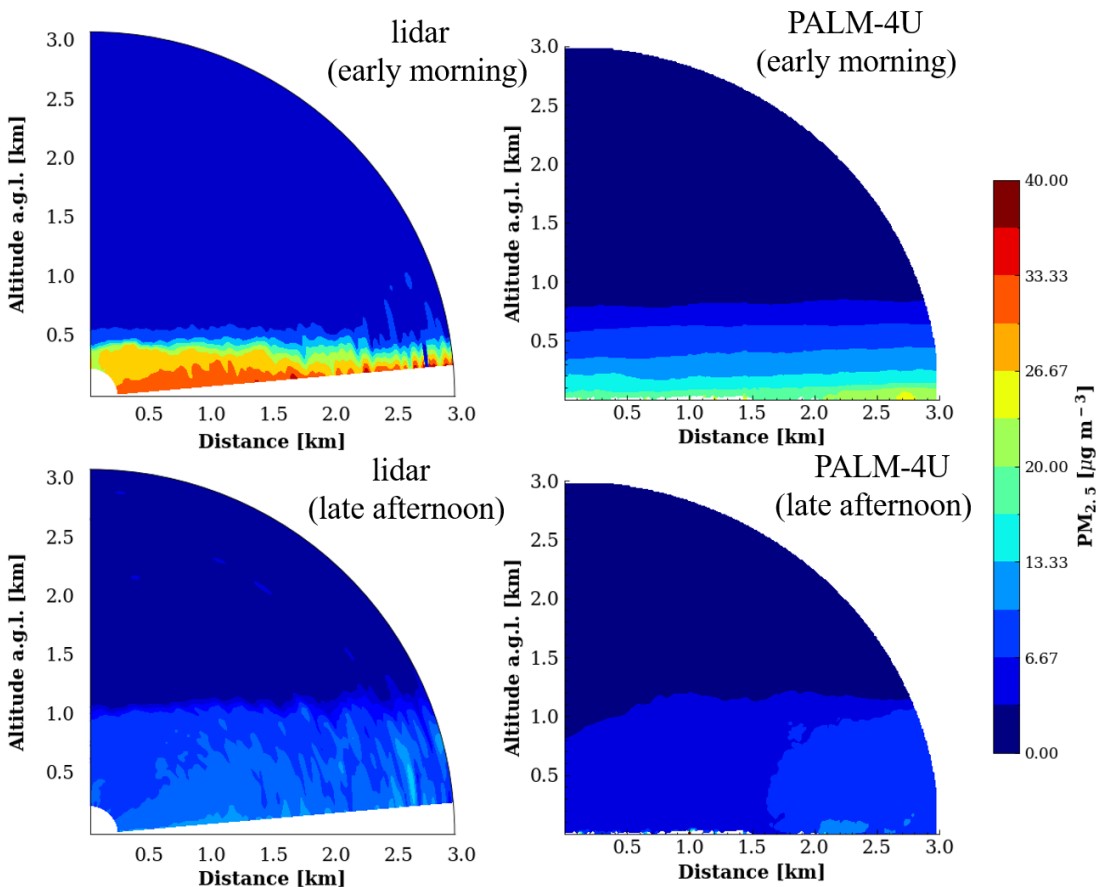

**Figure 9.** Range-height cross of $PM_{2.5}$ concentrations from scanning aerosol lidar (a, c) and PALM-4U simulation (b, d) at two different periods on February $14^{th}$, 2018. (a, b: 09:12 - 09:20; c, d: 16:07-16:15)

.

version factor calculated from ground-level $PM_{2.5}$ concentrations (OPC, Fidas200) and ground level lidar-derived extinction coefficients. Figure S9 shows a good linear correlation between extinction coefficients and $PM_{2.5}$ concentrations. Also, Figure 8c shows time series of $PM_{2.5}$ concentrations from ground level OPC measurements and from a PALM-4U simulation, which indicates that the simulated $PM_{2.5}$ concentrations show a similar trend as the observational data except for the spin-up period

(before 12:00 February $13^{th}$) but underestimate the $PM_{2.5}$ concentrations by a factor of $4.5 \pm 2.1$. The spin-up period ensures that the atmosphere is in balance with the new surface temperature and soil properties and that the atmospheric chemistry reaches an equilibrium state. The comparison of vertical extended $PM_{2.5}$ concentrations from lidar measurement and the model simulation shows that PALM-4U agrees well with lidar observation in terms of boundary layer evolution and aerosol mixing processes. However, the simulated average aerosol concentrations were lower than the observed concentrations by a factor of



$4.5 \pm 2.1$.

Several factors of uncertainty contribute to this underestimation, namely emissions as well as initial- and lateral boundary conditions (IC and LBC). Residential emissions are not included in this simulation, which we expect to be a significant source in winter. Furthermore, traffic emissions are parameterized based on street type and time of day (Khan et al., 2021). These very simple assumptions can not accurately simulate the true emissions, especially if traffic congestion amplifies true emissions.

High levels of HOA (traffic related aerosol) were identified in the PMF analysis of organic aerosol as shown in Figure 4c, which substantiates the assumption, that this is a large source of uncertainty in our simulation. Additionally, the train tracks approaching the Stuttgart main station are located near the lidar site, where break abrasion particulates are produced, which are also not simulated in the model. Finally, the IC and LBC for this simulations where based on the nocturnal profile at February 14 $^{th}$, 2018 00:00 (Here we used the data from February 14 $^{th}$, 2018 00:00 instead of February 13 $^{rd}$, 2018 00:00 be-

cause February 13 $^{rd}$, 2018 00:00 was cloudy, which would bring uncertainties in vertical aerosol profile retrieval.). While this proved to produce reasonable behaviour, providing spatially and temporally variable IC and LBC would most likely improve the agreement of the regional background concentrations and allow us to disentangle this contribution to the total uncertainty from the local emissions. It is also noteworthy, that only emissions, transport, and dry deposition were simulated here. Particulate processes like aggregation, wet deposition, chemical transformation or secondary aerosol formation are not accounted for,

with the underlying assumption being, that on local scales and 24 hour time scales, primary emissions and transport are more dominant.

Figure 9 shows range-height cross of PM$_{2.5}$ concentrations derived from lidar measurement and PLAM-4U simulation during two different periods (09:12 - 09:20; 16:07-16:15, UTC) on February 14$^{th}$, 2018. As we already know that the PALM-4U underestimated the PM$_{2.5}$ concentrations by a factor of $4.5 \pm 2.1$, we scaled up the PM$_{2.5}$ concentrations from PALM-4U sim-

ulation by a factor of 4.5 to better demonstrate aerosol spatial distribution. The figures on the first row demonstrate the aerosol spatial distribution in the early morning, which reflect a similar shallow boundary layer and high ground-level aerosol loading. The figures in the second row show the aerosol spatial distribution in the afternoon, which reflect a well mixing boundary layer and relatively low-concentration and homogeneous aerosol distributions. Compared with the observational data, PALM-4U simulated the aerosol spatial distribution and boundary layer structure well. However, there is still some inconsistency

that needs to be cleared. Compared to observational data, the model shows more homogeneous spatial and temporal aerosol distribution especially for the case in the afternoon as shown in the second row of figure 9. One possible reason for this inconsistency is that the PALM-4U did not resolve all turbulent eddies (i.e., limited by 10 m grid-spacing), and these unresolved eddies would contribute to spatial heterogeneity of aerosol. Another reason for this lack of structure could be associated with the time averaging 10 minutes might be long enough to blur many small scale instantaneous structures, that the instrument

might detect in a scan.




# 4   Conclusions

This study investigates boundary layer dynamics and air quality in a complex terrain by combining a scanning aerosol lidar, a wind lidar, a microwave radiometer, different *in-situ* aerosol characterization instruments, radiosondes, and a large eddy simulation for downtown Stuttgart in winter. The boundary layer heights retrieved from lidar show a good agreement with those from radiosondes with a slope of $1.102 \pm 0.135$ and a Pearson correlation coefficient of 0.86, respectively. This agreement reflects the good quality of our measurements and retrieval algorithms. Stagnant meteorological pattern with strong temperature inversion and low wind speeds can cause an accumulation of aerosol at ground level, contributing to significant air pollution events similar to previous observations in other cities (Jia et al., 2021; Li et al., 2021a; Huang et al., 2018).

Ground-level aerosol concentrations are anti-correlated with mixing layer heights but are correlated with stable boundary layer heights in the later night and early morning as reported by Yuval et al. (2020); Lou et al. (2019). The anti-correlation indicates that the convection within the boundary layer can dilute ground-level aerosol whereas the correlation means that this relationship is not only affected by boundary layer mixing process but also by local aerosol emissions (Huang et al., 2023; Tsai et al., 2011).

Two selected cases show that the evolution of boundary layer structures was affected by solar irradiation, clouds, temperature, as well as wind, and are greatly different under different meteorological conditions (Cao et al., 2020; Li et al., 2021b). Cloud cover during previous night time can significantly weaken the temperature inversion potentially causing a faster increase of boundary layer heights after sunrise. This is especially important for aerosol dilution during morning rush hours and demonstrates how strong different meteorological aspects influence air quality levels.

The comparison of PALM-4U model results with observational data shows that the simulated boundary layer dynamics and aerosol mixing processes are described relatively well by PALM-4U. However, it underestimates the $PM_{2.5}$ concentrations by a factor $4.5 \pm 2.1$. This underestimation is mainly due to uncertainties of emission as well as initial- and lateral boundary conditions (IC and LBC) (Khan et al., 2021; Maronga et al., 2020). Although the simulated aerosol concentrations are systematically lower than the observation values, the PALM-4U model still successfully reproduced the boundary layer evolution and its mixing effect on the ground level aerosol. This helps to better understand the boundary layer dynamics and the aerosol dispersion paths within the boundary layer.

PALM-4U model validation has been conducted at different places in terms of meteorological parameters as well as gas and particle pollutants (e.g. Oklahoma, USA; Münster, Germany; Prague, Czech Republic; Berlin, Germany; Christchurch, New Zealand; Hong Kong, China; Dresden, Germany) (Tewari et al., 2010; Paas et al., 2020; Resler et al., 2021; Khan et al., 2021; Lin et al., 2021; Wang et al., 2023). However, our research aims to contribute additional insights by focusing on the validation of boundary layer dynamics and aerosol mixing process within the boundary layer. This work presents one of the first validation of PLAM-4U in simulating aerosol distributions in a complex basin-like urban area. This study contributes to characterizing the structure of the urban boundary layer at a complex terrain and understanding the processes of air pollution in downtown Stuttgart. The impact of local emissions from different sources as well as horizontal and vertical transport can be distinguished based on this work. This is helpful to understand the influence of boundary layer mixing on aerosol evolution and to improve



air quality predictions and mitigation measures in urban areas with complex topography.

Furthermore, leveraging comprehensive observed data and high-resolution simulations from model outputs enables the reproduction of urban scenarios at the street level, which will contribute to advance the development of a digital twin for urban climates in the future (Chen et al., 2023; Schrotter and Hürzeler, 2020; Caprari et al., 2022).

*Code availability.* The code used to analyse the lidar data is property of Raymetrics Inc, but we have shown that it results in the same results
as the code single calculus chain (SCC) provided by EARLIENT (https://www.earlinet.org/index.php?id=earlinet_homepage, last access: 8 September 2023). The code of PLAM-4U model can be found in PALM website (https://palm.muk.uni-hannover.de/trac/wiki/palm4u, last access: 17 November, 2023).

*Data availability.* The lidar raw data and in situ measurement data are available via the open access data repository KIT open (link will be added). The radiosonde data are available upon request from the data originator (DWD; datenservice@dwd.de)

*Author contributions.* HS, XS, and WH performed the measurements. HZ analysed the scanning lidar data and combined all available data together. OK analyzed the data from radiosonde, wind lidar and temperature lidar. CH and BK performed and analysed the WRF- and PALM-4U simulations and wrote the corresponding parts of the manuscript. HZ wrote the manuscript with support from HS as well as contributions from all co-authors.

*Competing interests.* At least one of the co-authors is a member of the editorial board of Atmospheric Chemistry and Physics.

*Acknowledgements.* Support by the technical staff of the Institute of Meteorology and Climate Research and Germany weather service (DWD) especially Anderas Wieser and Bianca Adler for their support on wind lidar and microwave radiometer data collection. Financial support by the project Modular Observation Solutions for Earth Systems (MOSES) of the Helmholtz Association (HGF), finanical support by the German Federal Ministry of Education and Research (BMBF) under grants 01LP1602 G and 01LP1602 K within the research programme [UC][2]. Financial support by Horizon 2020 Framework Programme from European Research Council (CHAPAs, grant no. 850614). This work
was supported by the North-German Supercomputing Alliance (HLRN). We are grateful to the HLRN supercomputer staff, especially Stefan Wollny for his continual help and support.



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
