# Peer review of "Aerosol composition, air quality, and boundary layer dynamics in the urban background of Stuttgart in winter"

_EGUsphere, 2024_

## Referee Comment (RC1)

Review on Aerosol composition, air quality, and boundary layer dynamics in the urban background of Stuttgart in winter

The paper investigates the evolution of the planetary boundary layer (PBL) and associated aerosol concentrations for a field campaign and two case studies in winter 2018 in Stuttgart, Germany. The analysis uses a set of different in situ and remote sensing instruments to observe different characteristics of the PBL simultaneously. It further uses the PALM-4U model on a high spatial resolution (10m) to demonstrate the ability of such large eddy simulations to qualitatively reproduce the measured results. The study is in general well written and provide a meaningful contribution to the knowledge of inner-city PBL evolution and air quality.

Two aspects are my main concern. Firstly, it is not clear from the introduction to what extend the current publications differs from other investigations. E.g., the authors refer multiple times to Huang et al. (2019) but do not show the uniqueness of their own work. Secondly, the use of the PALM-4U model in this investigation is rather weak. There is a lot potential to use the model and validated conclusions made during the results section. It is rather added to this publication to show the overall agreement of the model with observations. I am certain that there are multiple publications that evaluate the PALM-4U model. Thus, I do not understand to what extent this paper differs from these previous investigations.

I recommend the author to carefully revise the manuscript and consider splitting it into two publications or extending the model evaluation to be more connected to the analysis made in the results section. Further details are given below.

General comments:

- The title suggests that the aerosol composition is investigated in the paper. However, the abstract does not account for this investigation. Please add this investigation to the abstract. Further, the abstract is short of quantitative results which can be of interest for other researchers.
- please provide a definition of "urban background in downtown Stuttgart"; As you mentioned in the text, the area of investigation in in-between a highly used road and the tracks heading to the main station. This does not sound like a "background" location.
- In many instances the paper refers to Huang et al. (2019). In the introduction there should be a paragraph that explains the differences to this study and how the current study complements the one by Huang et al.
- In the results section, the use of Figures is confusing and not consistent. I recommend to revise the paper and consider re-aranging the argumentation such that jumping back and forth between Figures will be avoided.
- The PALM-4U model plays a minor role in the analysis and does not contribute to the interpretation of the results. In fact, the observational data are used to validate, in a qualitative way, the PALM-4U simulation. Given the explanation in lines 461ff. is not sufficient for me. For a real investigation of PBL dynamics, a more thorough analysis would be required. I recommend to consider splitting the two parts (evaluation of PBL dynamics and air quality; verification of PALM-4U) to two publications or to use the model simulations to proof some of the results, e.g., the stability of the PBL in the case studies or regional transport effects. Further, I do not understand why the boundary conditions for the PALM-4U simulations are kept fixed. Please consider re-running the model with the improved boundary value handling to show also the effect of regional transport, vertical mixing of the aerosols on the different days etc.

Specific comments:
- line 1: Remove "the" in front of "cities"; further, add a comma "," before "which"
- line 10: "temporal-spatial" → "spatio-temporal"
- line 39: add "the" in front of "boundary layer"
- line 49: what do you mean by "Most lidars overlap"? Please clarify.
- line 54: add "the" in front of "nocturnal boundary layer"
- line 61: It is not possible to "compare" the PBL. Only characteristics of the PBL (e.g., PBL height) can be compared. Please clarify.
- line 55 – 64: I feel that this is not directly related to the paper as it focuses on a joint investigation between observations and LES. As the introduction is already long, I recommend to remove this part.
- line 65: I'd refer LES to be a method rather than a model. Please change.
- line 71: "qualified" does not seem to be the right word. I recommend to use "attribute" instead
- line 78: "strong" → "strongly"; further, explain the chemical "eBC" (also for other chemicals on first appearance)
- line 95: Correct the citation "Samad and Vogt" (year is missing)
- line 96: remove "of a city" or "in Stuttgart"
- Figure S1: Please exchange "left" and "right" in the caption
- line 105: Both, downtown and urban background, is used to describe the area of the field campaign. Please be more concise on the description. From my experience I'd not refer to Stuttgart Neckertor as urban background.
- Figure 1: Please use only one shading /contour color for the plot. Overlaying two colors makes it hard to distinguish the two displayed fields. I recommend to remove the temperature from the plot.
- line 123: you state that the valley floor is at approx 300 m but the Rosensteinpark is at 247 m. Please be more concise and correct the valley floor height.
- line 132: "a LES utilizing PLAM-4U" → I don't understand this phrase. Please clarify and correct the spelling of the model PALM-4U.
- line 150: description of variable "b" is missing. Please add. Also, $w_f$ is described as $W_f$ in the text. Please correct.
- line 188: please change "retrieved" to "retrieval"
- line 207: Please indicate the difference between PALM and PALM-4U. Please also consider restructuring the subsections. I recommend to rename subsection "Large eddy simulation" to "Modelling" or similar and but sections 2.4 an 2.5 as 2.3.1 and 2.3.2.
- line 216-218: This description of the initial profile that is taken as boundary profile is not clear to me. Please improve the description. Are you referring to the emission profile or the aerosol vertical profile?
- Figure 2: Change "Surfate" to "Sulfate" (Fig. 2b); The legend is not readable in Fig. 2e. In the caption, please correct the height of wind measurements (10m in the text, 2m in the caption).
- line 239: please provide the reason for the underestimation in the text.
- line 214/242: Your claim that the aerosol module of the model accounts for emissions, transport, deposition. From this, I derive that it does not account for chemical transformation. However, if a large fraction of aerosols is from ammonium nitrate (secondary aerosol), a discussion on the ability of the model to simulate the correct aerosol distribution is needed.
-line 258: This sentence needs more explanation. E.g., how can you derive stagnant dynamics from the shown profiles?
- line 262: If I interpret the scatter points correctly, two different regimes of PM10 concentrations are visible for PBL heights above 900m. Is this related to wind speed or transport effect from outside of the valley? Please give more details here
- Section 3.1: You use the term correlation throughout the section. However, this is rather empirical described by the plots. Please provide correlation values for all subplots of Fig. S5 to illustrate the strength of the correlation.

- line 305: please provide the dates for the two case studies.
- line 316: I find the reference to 1800 m confusing. What is the threshold of backscatter coefficients that lead to this conclusion? I'd rather claim that, in general, the aerosols are well mixed within the PBL (lidar) throughout the period
- line 317: at some point, the mentioned period should be explicitly dated.
- line223: "attitude" → "altitude"
- Figure S7: Please change the y-axis descriptions. In general, normalized values are unitless. Further, I do not understand the data you show. Shouldn't the different species concentrations add up to 100 % if you show normalized values? I assume you show staggered plots?
- line 332: please be more concise. Do you refer to Nitrate as "non-refractory particle mass" or where can I find this quantity in the plot?
- line 334-335: I would call a decrease of more than 50 % "slight". Please change.
- line 337-340: I am not sure if an increase of PBL height of 100 m can cause a reduction of eBC concentrations of more than 50 % but have no effect on other aerosols. I recommend to add some more details to this point.
- line 344: you mean Figure 4e? What do you mean by "and Figure 4 insert"?
- Caption Figure 5: "wavelength" -"wave"
- line 355: Add "UTC" after the time. Also for other places where the time is mentioned.
- line 379: "PLAM" → "PALM"
- line 380: Please rephrase and combine the two sentences to one.
- line 382-384: Do not go back to Figure 1 description. This is one example of confusing and non-consistent presentation of the results. I recommend to remove this statement as it is invaluable for the interpretation of the results.
- Figure 8: Why do you scale PM2.5 by a factor of 4 and not 4.5, which you have identified as bias between the model and observations?
- line 397: Again, you jump from Figure 8 to Figure 9 to Figure 8c within 3 sentences, without explicitly introducing Figure 9. Please restructure.
- Figure 9: What does the black line show? Is it arbitrary or does it reflect standard correlation values? Please provide explicit correlation values.
- line 404: Please remove as it is a repetition of results from above.
- line 414-415: please put this to the description of the PALM-4U model (around line 216, where I made another comment).
- line 416: Is there a proof for this statement, e.g., from the coarse resolution model runs?
- line 420: We saw in the Figures that nitrate is a significant contributor to local PM2.5 during this period. As nitrate is a secondary aerosol, how does this compares to your assumption, that secondary aerosols play a minor role?
- line 422: "cross" → "cross section"; "PLAM" → "PALM"
- line 453: Is it proofed that the local emissions are similar on the different days such that changes in emissions can be excluded as a possible source for the different aerosol concentrations? Please clarify.
- line 459: It is reasonable here to use the model to improve the understanding of the physical processes leading to the PBL evolution and to proof the findings from the analysis. See also my general comment on this aspect.

---

## Author Comment (AC1)

**Reply to the referee's comments**

We thank the referee for the useful comments, which helped us to improve the quality of our manuscript.

In the following, the referees' comments are given in black.

Our point-to-point replies are marked by "R" and are in blue.

Changes to the manuscript text are in green.

**Comments by referee #1**

The paper investigates the evolution of the planetary boundary layer (PBL) and associated aerosol concentrations for a field campaign and two case studies in winter 2018 in Stuttgart, Germany. The analysis uses a set of different in situ and remote sensing instruments to observe different characteristics of the PBL simultaneously. It further uses the PALM-4U model on a high spatial resolution (10m) to demonstrate the ability of such large eddy simulations to qualitatively reproduce the measured results. The study is in general well written and provide a meaningful contribution to the knowledge of inner-city PBL evolution and air quality. Two aspects are my main concern. Firstly, it is not clear from the introduction to what extend the current publications differs from other investigations. E.g., the authors refer multiple times to Huang et al. (2019) but do not show the uniqueness of their own work. Secondly, the use of the PALM-4U model in this investigation is rather weak. There is a lot potential to use the model and validated conclusions made during the results section. It is rather added to this publication to show the overall agreement of the model with observations. I am certain that there are multiple publications that evaluate the PALM4U model. Thus, I do not understand to what extent this paper differs from these previous investigations. I recommend the author to carefully revise the manuscript and consider splitting it into two publications or extending the model evaluation to be more connected to the analysis made in the results section. Further details are given below.

R: Thank you for your comments. This manuscript firstly investigates the boundary layer dynamics and associated aerosol concentrations as well as chemical compositions employing remote sensing, in-situ measurement as well as high-resolution model simulation, whereas most of previous publications used one or two of this methods (Kim & Kwon 2019, Zhang et al. 2020, Neff et al. 2008, Hennemuth & Lammert 2006, Froidevaux et al. 2013). This comprehensive datasets would contribute additional insights by focusing on the validation of boundary layer dynamics and aerosol mixing as well as transformation processes within the boundary layer.

We agree that the current utilization of the model is somewhat limited. Our initial aim was the validation of PALM-4U in simulating aerosol distributions in a complex basin-like urban area. This manuscript not only validated the boundary layer vertical mixing process but also aerosol spatial distributions. Furthermore, there is substantial potential of the PALM-4U model for deeper analysis and validation of our findings (e.g. The impact of clouds on boundary layer evolution and further on aerosol mixing and transformation processes). In future work, we plan to conduct more detailed comparison with model data based on better regional input, high resolution emission data as well as a more complete description of the physical and chemical transformation processes. Nonetheless, we think it is useful to demonstrate the actual model capabilities in comparison with this excellent observational data since there are on very few PALM-4U applications with aerosols so far.

To our best knowledge, this manuscript firstly used above comprehensive datasets to demonstrate boundary layer dynamics as well as aerosol mixing and transformation processes.

Also, we acknowledge that the current utilization of the model is somewhat limited and we are aware of the substantial potential of the PALM-4U model for a more detailed comparison with our comprehensive observations. For instance, we plan to conduct sensitivity tests on the impact of clouds on boundary layer evolution, on the aerosol mixing processes as well as aerosol physical and chemical transformation processes. Finally, in an upcoming study we aim to present more details of the model simulations in the context of different dynamic processes. Nonetheless, we think it is useful to demonstrate the actual model capabilities in comparison with this excellent observational data since there are on very few PALM-4U applications with aerosols so far.

**General comments:**

1. The title suggests that the aerosol composition is investigated in the paper. However, the abstract does not account for this investigation. Please add this investigation to the abstract. Further, the abstract is short of quantitative results which can be of interest for other researchers.

R: Thank you for your comments. We have modify the abstract accordingly as following (the change was highlighted in bold):

Aerosol distributions are of great relevance for air quality especially for cities like Stuttgart with limited air exchange due to its location in a basin. We collected a comprehensive set of data from remote sensing, *in-situ* methods including radiosondes for the urban background of downtown Stuttgart to determine the impact of boundary layer mixing processes on local air quality and to evaluate the simulation results of the **high-resolution large eddy simulation (LES) model PALM-4U at 10 m grid spacing**. Stagnant meteorological conditions caused accumulation of aerosols and chemical composition analysis shows that **ammonium nitrate ($37\% \pm 9\%$) and organic aerosol (OA, $34\% \pm 9\%$) dominated during this winter study**. Case studies show that clouds during previous nights can weaken temperature inversion and accelerate boundary layer mixing after sunrise **by up to 3 hours**. This is important for ground-level aerosol dilution during morning rush hours. Furthermore, our observations validate results of the LES model PALM-4U in terms of boundary layer heights and aerosol mixing for 48 hours. The simulated aerosol concentrations follow the trend of our observations but are still underestimated by a factor of $4.5 \pm 2.1$ due to missing secondary aerosol formation processes, uncertainties of emissions and boundary conditions in the model. This paper firstly evaluates the PALM-4U model performance in simulating aerosol spatio-temporal distributions, which can help to improve the LES model and to better understand sources and sinks for air pollution as well as the role of horizontal and vertical transport.

2. Please provide a definition of "urban background in downtown Stuttgart"; As you mentioned in the text, the area of investigation in in-between a highly used road and the tracks heading to the main station. This does not sound like a "background" location.

R: "Urban background" refers to a location within a city that is representative of typical urban conditions but is not directly influenced by specific, high-intensity sources of pollution, such as major roads or industrial sites. Our measurements were done in a park area in downtown Stuttgart with sufficient distance to heavy traffic or other substantial air pollution sources. There were no significant emissions from the electric train tacks nearby. Therefore, we can classify this indeed as an urban background site in a downtown area. We added the following sentences to the text.

Our measurements were done in a park area in downtown Stuttgart with sufficient distance to heavy traffic or other substantial air pollution sources. There were no significant emissions from the electric train tacks nearby. Therefore, we can classify this indeed as an urban background site in a downtown area.

3. In many instances the paper refers to Huang et al. (2019). In the introduction there should be a paragraph that explains the differences to this study and how the current study complements the one by Huang et al.

R: Huang et al. (2019) has reported the organic aerosol chemical composition and volatility for both winter and summer based data collected from the high-resolution time-of-flight aerosol mass spectrometer (HR-ToF-AMS) and filters samples analyzed by FIGAERO-CIMS. Their work provided insights into the seasonal variation of the molecular composition and volatility of ambient organic aerosol particles and into their potential sources. However, our work is more focused on the boundary layer evolution and associated aerosol spatial distribution within the boundary layer based on the comparison of the comprehensive dataset from remote sensing, *in-situ* measurements and model simulation.

Huang et al. (2019) has reported the organic aerosol chemical composition and volatility for both winter and summer, which provided insights into the seasonal variation of the molecular composition and volatility of ambient OA particles and into their potential sources. However, this work is more focused on the boundary layer evolution and associated aerosol spatial distribution within the boundary layer based on the comparison of the comprehensive dataset from remote sensing, *in-situ* measurements and model simulation.

4. In the results section, the use of Figures is confusing and not consistent. I recommend to revise the paper and consider re-arranging the argumentation such that jumping back and forth between Figures will be avoided.

R: We have re-arranged the argumentation and changed the figure positions accordingly to allow for a better flow. For example, we have moved the position of Figure 9 back by one page after to better fit the content.

5. The PALM-4U model plays a minor role in the analysis and does not contribute to the interpretation of the results. In fact, the observational data are used to validate, in a qualitative way, the PALM-4U simulation. Given the explanation in lines 461ff. is not sufficient for me. For a real investigation of PBL dynamics, a more thorough analysis would be required. I recommend to consider splitting the two parts (evaluation of PBL dynamics and air quality; verification of PALM-4U) to two publications or to use the model simulations to proof some of the results, e.g., the stability of the PBL in the case studies or regional transport effects. Further, I do not understand why the boundary conditions for the PALM4U simulations are kept fixed. Please consider re-running the model with the improved boundary value handling to show also the effect of regional transport, vertical mixing of the aerosols on the different days etc.

R: We agree that the current utilization of the model is somewhat limited. However, we have performed many versions of this simulation. Initially, we conducted simulations with cyclic boundary conditions (starting with constant profiles) and found the results to be sensitive to our initial profiles. We then proceeded with simulations using boundary conditions from the WRF model, as presented in the manuscript. Please note that the boundary conditions for the PALM-4U simulations were not fixed but were derived from the WRF simulation.

We have analyzed additional results from the PALM-4U simulation for this case. For example, we have wind data to examine aerosol transport and vertical thermal profiles to investigate boundary layer conditions. However, we consider these results to be beyond the scope of this manuscript. Christopher Claus Holst, one of the co-authors, will continue working on the model results for this case. Despite the limitations, we believe that the PALM-4U simulation remains valuable for this study, as it provides initial validation of PALM-4U in predicting aerosol spatial distributions in complex terrain. In an upcoming study we aim to present more details of the model simulations in the context of different dynamic processes.

More research is needed with more resources to address this in detail. Also, we acknowledge that the current utilization of the model is somewhat limited and we are aware of the substantial potential of the PALM-4U model for a more detailed comparison with our comprehensive observations. For instance, we plan to conduct sensitivity tests on the impact of clouds on boundary layer evolution, on the aerosol mixing processes as well as aerosol physical and chemical transformation processes. Finally, in an upcoming study we aim to present more details of the model simulations in the context of different dynamic processes. Nonetheless, we think it is useful to demonstrate the actual model capabilities in comparison with this excellent observational data since there are on very few PALM-4U applications with aerosols so far.

**Specific comments:**

1. line 1: Remove "the" in front of "cities"; further, add a comma "," before "which"

R: We have changed them.

2. line 10: "temporal-spatial" → "spatio-temporal".

R: We have changed it.

3. line 39: add "the" in front of "boundary layer".

R: We have added "the" in front of "boundary layer".

4. line 49: what do you mean by "Most lidars overlap"? Please clarify.

R: The overlap is different from different lidar system, but most of them range from tens of meters to around one thousand meters. To make it clear, we change this sentence as following:

However, most lidars provide interpretable data at distances from tens of meters to around one thousand meters, which makes it difficult to get valid measurements near the surface level for most vertically pointing lidar system.

5. line 54: add "the" in front of "nocturnal boundary layer"

R: We have added "the" in front of "nocturnal boundary layer".

6. Iline 61: It is not possible to "compare" the PBL. Only characteristics of the PBL (e.g., PBL height) can be compared. Please clarify.

R: We have removed this part according to comment 7.

7. line 55 – 64: I feel that this is not directly related to the paper as it focuses on a joint investigation between observations and LES. As the introduction is already long, I recommend to remove this part.

R: we have removed this part.

8. line 65: I'd refer LES to be a method rather than a model. Please change.

R: We have changed "model" to "method".

9. line 71: "qualified" does not seem to be the right word. I recommend to use "attribute" instead

R: We have changed "qualified" to "attribute".

10. line 78: "strong" → "strongly"; further, explain the chemical "eBC" (also for other chemicals on first appearance)

R: We have changed "strong" to "strongly". We also added "equivalent Black Carbon (eBC)".

11. line 95: Correct the citation "Samad and Vogt" (year is missing)

R: We have changed the format of this citation.

12. line 96: remove "of a city" or "in Stuttgart"

R: We have removed "of a city".

13. Figure S1: Please exchange "left" and "right" in the caption.

R: We have exchanged "left" and "right" in the caption of Figure S1.

14. line 105: Both, downtown and urban background, is used to describe the area of the field campaign. Please be more concise on the description. From my experience I'd not refer to Stuttgart Neckertor as urban background.

R: "Urban background" refers to a location within a city that is representative of typical urban conditions but is not directly influenced by specific, high-intensity sources of pollution, such as major roads or industrial sites. Our measurements were done in a park area in downtown Stuttgart with sufficient distance to heavy traffic or other substantial air pollution sources. There were no significant emissions from the electric train tacks nearby. Therefore, we can classify this indeed as an urban background site in a downtown area. We added the following sentences to the text.

Our measurements were done in a park area in downtown Stuttgart with sufficient distance to heavy traffic or other substantial air pollution sources. There were no significant emissions from the electric train tacks nearby. Therefore, we can classify this indeed as an urban background site in a downtown area.

15. Figure 1: Please use only one shading /contour color for the plot. Overlaying two colors makes it hard to distinguish the two displayed fields. I recommend to remove the temperature from the plot.

R: Thank you for the comment. This figure uses perceptually uniform colormaps (cet): isoluminant color for temperature and linear brightness for topography. While on printed paper it looks unclear, on screen it is very readable for most people. The exception is a rare color blindness where red color looks grey. Presenting the relationship between local topography height and local temperature distribution is quite challenging and much effort went into calibrating the exact lightness and transparency of the color overlay and the linear greyscale for good readability on screen. Thus we choose to not adjust the figure to reduce its shown information.

16. line 123: you state that the valley floor is at approx 300 m but the Rosensteinpark is at 247 m. Please be more concise and correct the valley floor height.

R: We think this statement is still valid as the valley floor covers a broader area and the Rosensteinpark is only a point. So we give an approximate number for an average valley altitude but an exact value for the measurement location at the Rosensteinpark.

17. line 132: "a LES utilizing PLAM-4U" → I don't understand this phrase. Please clarify and correct the spelling of the model PALM-4U.

R: We have changed to "an LES applying PLAM-4U".

Furthermore, an LES applying PALM-4U (Maronga et al. 2020) was performed to simulate the complex airflow and resulting aerosol transport in this area.

18. line 150: description of variable "b" is missing. Please add. Also, wf is described as Wf in the text. Please correct.

R: We have added "b is the translation parameter" and changed "Wf" to "wf"

The dilation a is set to be 75 m in this paper and b is the translation parameter.

19. line 188: please change "retrieved" to "retrieval".

R: We have changed "retrieved" to "retrieval"

20. line 207: Please indicate the difference between PALM and PALM-4U. Please also consider restructuring the subsections. I recommend to rename subsection "Large eddy simulation" to "Modelling" or similar and but sections 2.4 an 2.5 as 2.3.1 and 2.3.2.

R: PALM-4U (Maronga et al. 2020) is a model system that has been developed to simulate a wide range of urban micro-scale processes. The center of this model system is the large eddy simulation model PALM (Raasch & Schröter 2001) based on non-hydrostatic, filtered, incompressible Navier-Stokes equations in Boussinesq-approximated form. We have added this as follows:

PALM-4U (Maronga et al. 2020) is a model system that has been developed to simulate a wide range of urban micro-scale processes. The center of this model system is the large eddy simulation model PALM (Raasch & Schröter 2001) based on non-hydrostatic, filtered, incompressible Navier-Stokes equations in Boussinesq-approximated form.

21. line 216-218: This description of the initial profile that is taken as boundary profile is not clear to me. Please improve the description. Are you referring to the emission profile or the aerosol vertical profile?

R: For the PALM-4U simulation run, we need the initial boundary conditions and later boundary conditions. For this simulation, we used aerosol vertical profiles from lidar observations as initial boundary conditions and WRF output as later boundary conditions. And we also need sources and sinks for aerosol simulation. In this simulation, the emissions sources of the model were parameterized by street types. To explain this we have modified the text as follows:

The emissions sources of the PALM-4U model were parameterized by street types (Maronga et al. 2020) and initial boundary conditions profiles were approximated from observed profile values at simulation initialization time.

22. Figure 2: Change "Surfate" to "Sulfate" (Fig. 2b); The legend is not readable in Fig. 2e. In the caption, please correct the height of wind measurements (10m in the text, 2m in the caption).

R: We have modified Figure 2 accordingly. The wind speed was measured at 10 m above ground and we changed the caption.

[Figure]

Figure 2: Time series of range corrected lidar signal (contour) and boundary layer heights derived from scanning aerosol lidar (pink line), radiosonde (stars), and ERA5 dataset (black dashed line) (a), the aerosol mass concentrations for different chemical components (b), five-factor PMF solutions of organic aerosol (c), the particle matter concentrations measured by OPC (d), and the temperatures at two different altitude levels measured by radiosonde as well as wind speed measured at 10 m above the ground level (e).

23. line 239: please provide the reason for the underestimation in the text.

R: We believe that the differences between observations and ERA5 boundary layer heights is mainly due to the different spatial resolutions of the methods and the relatively complex topography in Stuttgart.

The inconsistency between observation and ERA5 mainly due to the different spatial resolutions of the methods and the relatively complex topography in Stuttgart.

24. line 214/242: Your claim that the aerosol module of the model accounts for emissions, transport, deposition. From this, I derive that it does not account for chemical transformation. However, if a large fraction of aerosols is from ammonium nitrate (secondary aerosol), a discussion on the ability of the model to simulate the correct aerosol distribution is needed.

R: Thank you for pointing this out. Actually, this model version did not consider the aerosol chemical composition and only $PM_{2.5}$ and $PM_{10}$ were predicted via prognostic scalar transport equations. Hence the formation of secondary aerosol generated by chemical reactions is not considered in this work and it would be the next step of our work to include this. In scope of this study we did not have computational resources and manpower to set up and test a full SALSA aerosol physics simulation. Given that we attempted the first winter evaluation of PALM's aerosol simulation behavior in complex urban terrain, our objective was to mostly check for plausible boundary layer dynamics and spatial patterns. More research is needed with more resources to address this in detail. We added several clarifications into the text to specify the limitations and our objective for this simulation more precisely.

Regarding future work related to this study, this model version did not consider the aerosol chemical composition and only $PM_{2.5}$ and $PM_{10}$ were predicted via prognostic scalar transport equations. Hence the formation of secondary aerosol generated by chemical reactions is not considered in this work and it would be the next step of our work to include this. In scope of this study we did not have computational resources and manpower to set up and test a full SALSA aerosol physics simulation. Given that we attempted the first winter evaluation of PALM's aerosol simulation behavior in complex urban terrain, our objective was to mostly check for plausible boundary layer dynamics and spatial patterns. More research is needed with more resources to address this in detail.

25. line 258: This sentence needs more explanation. E.g., how can you derive stagnant dynamics from the shown profiles?

R: We have added following sentence to manuscript.

Figure S4 shows a strong temperature inversion and low wind speed during a polluted period, which is the typical vertical thermal and dynamic structure during stagnant conditions (Huang et al. 2018).

26. line 262: If I interpret the scatter points correctly, two different regimes of PM10 concentrations are visible for PBL heights above 900m. Is this related to wind speed or transport effect from outside of the valley? Please give more details here.

R: As the wind speed is $1.18 \pm 0.62$ m/s (less than 2 m/s), the aerosol horizontal transport could be neglected in first approximation. The reason for aerosol above 900 m is due to the boundary layer mixing process e.g. local vertical updraft. The aerosol was diluted by transporting them from near ground level to higher altitudes during boundary layer mixing .

The aerosol was diluted by transporting them from near ground level to higher altitudes during boundary layer mixing.

27. Section 3.1: You use the term correlation throughout the section. However, this is rather empirical described by the plots. Please provide correlation values for all subplots of Fig. S5 to illustrate the strength of the correlation.

R: As the correlation for each subplots is not consistent (e.g., Figure S5 shows that $PM_{10}$ is correlated with boundary layer below 900 m but is anti-correlated with boundary layer above 900 m). It would be complex to give each correlation coefficient in the figure. Therefore, we added the correlation coefficients in the text when we mention the respective correlation.

Figure S5 (a, e, i) shows the correlation between boundary layer heights and $PM_{10}$, eBC, as well as BBOA concentrations for three different subsets of data, respectively. The color of the scatter points indicates the relative humidity. For all $PM_{10}$ data points an anti-correlation as shown in figure S5a, was found for boundary layer heights above 900 m (R = -0.44, pearson correlation coefficient, same hereafter). This anti-correlation means that a deeper boundary layer diluted the aerosol while a shallower boundary layer concentrated aerosol at the ground level. The aerosol was diluted by transporting them from near ground level to higher altitudes during boundary layer mixing process. However, we also found a positive correlation between $PM_{10}$ and boundary layer heights for boundary layer heights below 900 m (a.s.l.) (R = 0.32). This positive correlation is also reported in Yuval et al. (2020) and typically coincided with low wind speed and high relative humidity, indicating typical properties of the nocturnal boundary layers.
Then the data was divided into three groups for three different time periods - morning (04:00 - 10:00, UTC) (b, f, j), afternoon (12:00-18:00 UTC) (c, g, k), and night (18:00 - 04:00, UTC) (d, h, l). The correlation between the boundary layer and surface aerosol concentrations ($PM_{10}$) in the these three subplots (b, c, d) show a positive correlation for PBL heights below 900 m (a.s.l.) (R = 0.31 (figure S5b), 0.58 (figure S5ad)) and a weaker but negative correlation for larger PBL heights (R = -0.64(figure S5b), -0.49 (figure S5c), -0.22 (figure S5d)). The correlation between boundary layer heights and eBC as well as BBOA concentrations shown in Figure S5 revealed that the eBC and BBOA concentrations are always anti-correlated with the boundary layer heights (R = -0.25 (figure S5e), 0.21 (figure S5i)). The reason for the positive correlation between $PM_{10}$ and boundary layer height below 900 m a.s.l. is due to the local emissions and aerosol water take up during night and early morning. The reason for only anti-correlation between the boundary layer heights and eBC as well as BBOA concentrations is that the eBC and BBOA particles emission from sources like biomass burning or traffic are smaller and less hygroscopic and thus could be diluted by boundary layer evolution.

28. line 305: please provide the dates for the two case studies.

R: We have added the date for the two cases (February $13^{rd}$ - February $14^{th}$, 2018 and February $24^{th}$ - February $25^{th}$, 2018).

29. line 316: I find the reference to 1800 m confusing. What is the threshold of backscatter coefficients that lead to this conclusion? I'd rather claim that, in general, the aerosols are well mixed within the PBL (lidar) throughout the period.

R: Actually, 1800 m is the maximum of boundary layer height for these two days. We changed the sentence as follows:

The vertically extended backscatter coefficients in this figure show that most of the aerosol only stayed within the boundary layer or residual layer and could not reach to the free troposphere as stated in previous publications (Guo et al. 2009, Quan et al. 2013, Li et al. 2017, Su et al. 2018, Yuval et al. 2020).

30. line 317: at some point, the mentioned period should be explicitly dated.

R: We have removed "whole period" to clarify this sentence following the comment 29.

31. line223: "attitude" $\rightarrow$ "altitude"

R: We have changed "attitude" to "altitude"

32. Figure S7: Please change the y-axis descriptions. In general, normalized values are unitless. Further, I do not understand the data you show. Shouldn't the different species concentrations add up to 100 % if you show normalized values? I assume you show staggered plots?

R: We would like to clarify our normalization process. The normalization means the aerosol concentration was normalized by boundary height in the two following steps: (1) The boundary layer height was normalized to get unit-less boundary layer height (2) We multiply the aerosol concentration by the unit-less boundary layer height. Hence, the unit is correct. We have added the following sentence to the manuscript.

In order to investigate the effect of local emission on ground level aerosol concentration, we normalized the aerosol concentrations by the boundary layer heights. The normalization was conducted in two subsequent steps: (1) The boundary layer height was normalized to get a unit-less boundary layer height. (2) We multiplied the aerosol concentrations by the unit-less boundary layer heights.

33. line 332: please be more concise. Do you refer to Nitrate as "non-refractory particle mass" or where can I find this quantity in the plot?

R: All the chemical components measured by the aerosol mass spectrometer (AMS) refer to non-refractory particles (e.g. everything evaporating at 600°C). We have clarified this in the manuscript by changing this sentence as following.

This figure shows that the nitrate aerosol particle mass increased from 3.9 $\mu$g/m$^3$ to 10.8 $\mu$g/m$^3$ at morning rush hours (06:00 -12:00, UTC) on February $14^{th}$, 2018.

34. line 334-335: I would call a decrease of more than 50 % "slight". Please change.

R: We have changed "a slight decrease" into a decrease of more than 50 % .

While during the night time (18:00 - 04:00, UTC), the nitrate aerosol particle mass increased from 2.5 $\mu$g/m$^3$ to 3.9 $\mu$g/m$^3$, the eBC concentrations decreased by more than 50 % from 1048 ng/m$^3$ to 464 ng/m$^3$.

line 337-340: 35. I am not sure if an increase of PBL height of 100 m can cause a reduction of eBC concentrations of more than 50 % but have no effect on other aerosols. I recommend to add some more details to this point.

R: The eBC concentrations have been normalized by the boundary layer height (day time) or residual layer height (night time). This is not the aerosol concentration but the normalized value. As you can see, the eBc concentrations shown in Figure 4 are almost constant from 18:00, February $13^{rd}$ to 04:00 February $13^{th}$, UTC. The normalized value decreased during this time due to the decrease of the residual layer height. We have added following sentences to the manuscript to explain this.

However, we need to be careful with this result especially during night time as the aerosols are not well mixed but accumulated near ground level. Hence, this normalization would lead to an underestimation when

considered as total aerosol concentration within the boundary layer.

36. line 344: you mean Figure 4e? What do you mean by "and Figure 4 insert"?

R: Sorry for this mistake. We have changed "Figure 4d" to "Figure 4e" and changed "Figure 4 insert" to "Figure 4 h". Figure 4 was also changed accordingly.

[Figure]

Figure 4: Time series of backscatter coefficients from lidar measurements (contour plot), the boundary layer heights from lidar measurement (white solid line), the ERA5 dataset (grey dashed line), and DWD radiosonde (yellow triangle) as well as residual layer heights retrieved from lidar (white dashed line) (a), the aerosol mass concentrations measured by aerosol mass spectrometer (AMS) (b), five-factor positive matrix factorization (PMF) solutions of organic aerosol(c), black carbon concentrations (d), potential temperature measured by microwave radiometer (MWR) (e), turbulence kinetic energy (TKE) retrieved from Doppler lidar (f) as well as the global radiation measured by meteorological sensors (WS700) (g) for case 1 from February $13^{th}$ to February $14^{th}$, 2018. The white arrows in panel a and b show the decreasing or increasing trends of boundary layer height. The plot on the right side shows the potential temperatures measured by radiosonde at 06:00 of $13^{th}$ and $14^{th}$, February, 2018.

.

37. Caption Figure 5: "wavelength" -"wave"

R: We have changed "wavelength" to "wave".

38. line 355: Add "UTC" after the time. Also for other places where the time is mentioned.

R: We have added "UTC" after the times.

39. line 379: "PLAM" → "PALM"

R: We have changed "PLAM" to "PALM".

40. line 380: Please rephrase and combine the two sentences to one.

R: We have changed these two sentences to one as following:

To simplify, we use the altitude above the ground level (a.g.l.) to compare observational results with model simulations as the coordinate used in the model is the height above ground level.

41. line 382-384: Do not go back to Figure 1 description. This is one example of confusing and nonconsistent presentation of the results. I recommend to remove this statement as it is invaluable for the interpretation of the results.

R: We have removed this statement.

42. Figure 8: Why do you scale PM2.5 by a factor of 4 and not 4.5, which you have identified as bias between the model and observations?

R: We agree and modified Figure 8 accordingly.

43. line 397: Again, you jump from Figure 8 to Figure 9 to Figure 8c within 3 sentences, without explicitly introducing Figure 9. Please restructure.

R: We have restructure this part as follows:

To compare aerosol spatial concentrations between the PALM-4U simulation and observations, we converted lidar-derived extinction coefficients to $PM_{2.5}$ concentrations using a conversion factor calculated from ground-level $PM_{2.5}$ concentrations (OPC, Fidas200) and ground-level lidar-derived extinction coefficients. Figure S9 shows a good linear correlation between extinction coefficients and $PM_{2.5}$ concentrations with slop of $78182.0 \pm 1132.0$ and Person correlation coefficient of 0.822, and this good correlation ensures the quality of this conversion. Figure 8 shows the time series of the $PM_{2.5}$ concentrations retrieved from lidar measurements and PALM-4U.

44. Figure 9: What does the black line show? Is it arbitrary or does it reflect standard correlation values? Please provide explicit correlation values.

R: The black line in figure S9 is a linear fit and we added the correlation coefficient to the figure .

[Figure]

Figure S9: Correlation of ground-level extinction coefficients from lidar retrieval and $PM_{10}$ concentrations from Fidas200 measurements from February $5^{th}$ to March $5^{th}$, 2018 in Stuttgart. The linear fit shown as black line has a correlation coefficient of $78182.0 \pm 1132.0$.

45. line 404: Please remove as it is a repetition of results from above.

R: We have removed this sentence.

46. line 414-415: please put this to the description of the PALM-4U model (around line 216, where I made another comment).

R: We have put this to the description of the PALM-4U model section and modified as following in order to better fit into that section.

...initial boundary conditions profiles approximated from observed profile values at simulation initialization time.

47.line 416: Is there a proof for this statement, e.g., from the coarse resolution model runs?

R: We did investigate the parent domain outputs and found that the large distance from outer domain boundary to inner domain boundary (> 3BLH during daytime) allowed sufficient "spin-up" mixing upstream of the child domain, such that turbulence and vertical distribution behaved plausibly. Depending on the grid spacing of 40 m, the spatial heterogeneity transported into the child domain was however quite diffuse, as expected. This point was added into the manuscript. We also rephrased the sentence, as we agree that the chosen form was not adequate.

We found that the large distance from outer domain boundary to inner domain boundary (> 3BLH during daytime) allowed sufficient "spin-up" mixing upstream of the child domain, such that turbulence and vertical distribution behaved plausibly. Depending on the grid spacing of 40 m, the spatial heterogeneity transported into the child domain was however quite diffuse, as expected.

48. line 420: We saw in the Figures that nitrate is a significant contributor to local PM2.5 during this period. As nitrate is a secondary aerosol, how does this compares to your assumption, that secondary aerosols play a minor role?

This is indeed an important point. We added a brief discussion about this into the manuscript. According to LUBW analysis the annual average traffic contribution is 58% to the total PM. As we do parameterize road emissions, this contribution is partially accounted for. Noteworthy, though the parent domain with a grid spacing of 40 m cannot fully represent all smaller roads, thus we expect an underestimation in the "urban regional" road emissions outside the child domain. In addition to that, the annual average does not reflect the winter conditions, during which residential heating would have contributed significantly to the total particulate emissions. When adding up both of these underestimations, we assume, that the underestimation caused by ignoring secondary aerosol formation is most likely smaller. As we did not have access to emission inventories with sufficiently high resolution (sub 100m) and granularity of aerosol properties, it was not feasible to attempt a full evaluation of aerosol process properties using SALSA, as we did not have dedicated resources in personnel or compute time for this experiment and thus focused on boundary layer dynamics, which we did have the tools to evaluate properly. We understand that this requires more research in the future, but we believe that in the scope of this manuscript, the objective to test for correct boundary layer dynamics and the associated mixing and transport behavior, we contributed an important first step.

It is also noteworthy, that only road emissions, transport, and dry deposition were simulated here. Particulate processes like aggregation, wet deposition, chemical transformation or secondary aerosol formation are not accounted for, with the underlying assumption being, that on local scales and 24 hour time scales, primary emissions and transport are more dominant. As mentioned in the introduction, 58 % of the concentrations have been attributed to road emissions (LUBW 2019). This being an annual average,the omission of residential heating emissions most likely accounts for a large fraction of the simulated underestimation during this cold winter period. Additionally, most smaller roads are not fully represented in the parent domain with 40m grid spacing, such that the regional urban background road emissions might be underrepresented with contributions only from highways and other large road structures. This is substantiated by our finding, that HOA is dominant in these periods as shown in Figure 4d.

49. line 422: "cross" → "cross section"; "PLAM" → "PALM"

R: We have changed "cross" to "cross section" annd "PLAM" to "PALM".

50. line 453: Is it proofed that the local emissions are similar on the different days such that changes in emissions

can be excluded as a possible source for the different aerosol concentrations? Please clarify.

R: Because the two days choosen (February $13^{th}$ and February $14^{th}$) were on weekdays, the changes in emissions can be excluded. Hence the different ground aerosol concentrations on these two days are mainly due to different boundary layer mixing.

The different boundary layer mixing on these two day has a substantial impact on ground level aerosol concentrations. As can be seen from 4, significantly more aerosol accumulated on February $14^{th}$ due to lower boundary layer heights before 12:00, UTC. Here we assume similar emissions on these two weekdays.

51. line 459: It is reasonable here to use the model to improve the understanding of the physical processes leading to the PBL evolution and to proof the findings from the analysis. See also my general comment on this aspect.

R: Thank you for your comments.

**References**

Froidevaux, M., Higgins, C. W., Simeonov, V., Ristori, P., Pardyjak, E., Serikov, I., Calhoun, R., van den Bergh, H. & Parlange, M. B. (2013), 'A raman lidar to measure water vapor in the atmospheric boundary layer', *Advances in Water Resources* **51**, 345–356. 35th Year Anniversary Issue.
**URL:** *https://www.sciencedirect.com/science/article/pii/S0309170812000929*

Guo, J.-P., Zhang, X.-Y., Che, H.-Z., Gong, S.-L., An, X., Cao, C.-X., Guang, J., Zhang, H., Wang, Y.-Q., Zhang, X.-C., Xue, M. & Li, X.-W. (2009), 'Correlation between pm concentrations and aerosol optical depth in eastern china', *Atmospheric Environment* **43**(37), 5876–5886.
**URL:** *https://www.sciencedirect.com/science/article/pii/S1352231009007493*

Hennemuth, B. & Lammert, A. (2006), 'Determination of the atmospheric boundary layer height from radiosonde and lidar backscatter', *Boundary-Layer Meteorology* **120**(1), 181–200.

Huang, Q., Cai, X., Wang, J., Song, Y. & Zhu, T. (2018), 'Climatological study of the boundary-layer air stagnation index for china and its relationship with air pollution', *Atmospheric Chemistry and Physics* **18**(10), 7573–7593.
**URL:** *https://acp.copernicus.org/articles/18/7573/2018/*

Huang, W., Saathoff, H., Shen, X., Ramisetty, R., Leisner, T. & Mohr, C. (2019), 'Seasonal characteristics of organic aerosol chemical composition and volatility in stuttgart, germany', *Atmospheric Chemistry and Physics* **19**(18), 11687–11700.
**URL:** *https://acp.copernicus.org/articles/19/11687/2019/*

Kim, M.-S. & Kwon, B. H. (2019), 'Estimation of sensible heat flux and atmospheric boundary layer height using an unmanned aerial vehicle', *Atmosphere* **10**(7), 363.

Li, Z., Guo, J., Ding, A., Liao, H., Liu, J., Sun, Y., Wang, T., Xue, H., Zhang, H. & Zhu, B. (2017), 'Aerosol and boundary-layer interactions and impact on air quality', *National Science Review* **4**(6), 810–833.
**URL:** *https://doi.org/10.1093/nsr/nwx117*

LUBW (2019), 'Luftreinhaltepläne für Baden-Württemberg'.
**URL:** *https://pudi.lubw.de/detailseite/-/publication/37937*

Maronga, B., Banzhaf, S., Burmeister, C., Esch, T., Forkel, R., Fröhlich, D., Fuka, V., Gehrke, K. F., Geletič, J., Giersch, S., Gronemeier, T., Groß, G., Heldens, W., Hellsten, A., Hoffmann, F., Inagaki, A., Kadasch, E., Kanani-Sühring, F., Ketelsen, K., Khan, B. A., Knigge, C., Knoop, H., Krč, P., Kurppa, M., Maamari, H., Matzarakis, A., Mauder, M., Pallasch, M., Pavlik, D., Pfafferott, J., Resler, J., Rissmann, S., Russo, E., Salim, M., Schrempf, M., Schwenkel, J., Seckmeyer, G., Schubert, S., Sühring, M., von Tils, R., Vollmer, L., Ward, S., Witha, B., Wurps, H., Zeidler, J. & Raasch, S. (2020), 'Overview of the palm model system 6.0', *Geoscientific Model Development* **13**(3), 1335–1372.
**URL:** *https://gmd.copernicus.org/articles/13/1335/2020/*

Neff, W., Helmig, D., Grachev, A. & Davis, D. (2008), 'A study of boundary layer behavior associated with high no concentrations at the south pole using a minisodar, tethered balloon, and sonic anemometer', *Atmospheric Environment* **42**(12), 2762–2779. Antarctic Tropospheric Chemistry Investigation (ANTCI) 2003.
**URL:** *https://www.sciencedirect.com/science/article/pii/S1352231007001057*

Quan, J., Gao, Y., Zhang, Q., Tie, X., Cao, J., Han, S., Meng, J., Chen, P. & Zhao, D. (2013), 'Evolution of planetary boundary layer under different weather conditions, and its impact on aerosol concentrations',

*Particuology* **11**(1), 34–40. Recent Advances for Aerosol and Environment Study in Asia.
**URL:** *https://www.sciencedirect.com/science/article/pii/S1674200112001137*

Raasch, S. & Schröter, M. (2001), 'Palm - a large-eddy simulation model performing on massively parallel computers', *Meteorologische Zeitschrift* **10**(5), 363–372.
**URL:** *http://dx.doi.org/10.1127/0941-2948/2001/0010-0363*

Su, T., Li, Z. & Kahn, R. (2018), 'Relationships between the planetary boundary layer height and surface pollutants derived from lidar observations over china: regional pattern and influencing factors', *Atmospheric Chemistry and Physics* **18**(21), 15921–15935.
**URL:** *https://acp.copernicus.org/articles/18/15921/2018/*

Yuval, Levi, Y., Dayan, U., Levy, I. & Broday, D. M. (2020), 'On the association between characteristics of the atmospheric boundary layer and air pollution concentrations', *Atmospheric Research* **231**, 104675.
**URL:** *https://www.sciencedirect.com/science/article/pii/S0169809519302819*

Zhang, Y., Wang, L., Santanello, J. A., Pan, Z., Gao, Z. & Li, D. (2020), 'Aircraft observed diurnal variations of the planetary boundary layer under heat waves', *Atmospheric Research* **235**, 104801.
**URL:** *https://www.sciencedirect.com/science/article/pii/S0169809519311305*

---

## Author Comment (AC2)

**Reply to the referee's comments**

We thank the referee for the useful comments, which helped us to improve the quality of our manuscript.

In the following, the referees' comments are given in black.

Our point-to-point replies are marked by "R" and are in blue.

Changes to the manuscript text are in green.

**Comments by referee #2**

This is a quite complete paper about the evolution of the planetary boundary layer and its relationship with the recorded concentrations at a polluted site in Germany. Stuttgart was the selected site and the period investigated extended for February $5^{th}$ to March $5^{th}$ in 2018. Although this period is short, the equipment used is formed by a scanning aerosol lidar, an aerosol mass spectrometer, an aethalometer, a condensation particle counter, an optical particle counter, trace gas sensors and meteorological sensors. Measurements of these devices are presented. Moreover, experimental observations are contrasted with model simulations. The paper focused on the planetary boundary layer evolution during certain days and the role played by clouds. Additionally, a longer database, which extended form January 1st 2020 to January 1st 2022, was used to present the daily evolution of PM10 concentrations and the boundary layer height. Consequently, the paper merits to be published in Atmospheric Chemistry and Physics after the introduction of the following minor changes.

Since the period investigated is short, only one month, the results representativeness could be questioned. Hence, the authors should indicate if their results are robust enough.

R: Although the investigated period is short, we collected a comprehensive dataset which allow us better to understand the evolution of the planetary boundary layer (PBL) and associated aerosol concentrations through case studies. Furthermore, the meteorological conditions during the measurement period can be considered as quite typical winter conditions under high-pressure system influence. In addition, the correlation between boundary layer and aerosol distribution revealed by this short-term dataset fitted well with a 2-years dataset, which supports the robustness of our results. Finally, we need to be careful about the uniqueness of our result. As the measurement site located at a special topography in a step valley, the special topography may have impact on the boundary layer evolution and on accumulation of ground level aerosol concentrations. This is also the reason why we chose Stuttgart to investigate this topic.

Although the investigated time period is relatively short, the correlation between boundary layer and aerosol distribution revealed by this dataset fitted well with a 2-year dataset, which supports the robustness of our results. Furthermore, the meteorological conditions during the measurement period can be considered as quite typical winter conditions under high-pressure system influence. Therefore, our results have sufficient representativeness to compare with e.g. other seasons.

Moreover, some orthographic features are in the surroundings. The authors should comment the influence of such features on the planetary boundary layer evolution following varied wind directions. A short comment about possible processes such as anabatic or katabatic winds would increase the value of this paper.

R: Thank you for pointing on this. Actually, we have seen the phenomenon that anabatic or katabatic winds have impact on the boundary layer height. We have added the following text into manuscript to point this out.

Furthermore, the boundary layer heights measured by radiosonde is higher than those derived from lidar measurements, in contrast to case 1 (Figure 4). The possible reason for the differences in boundary layer height for case 2 can be explained as follows: the radiosonde site (SB, 321 m a.s.l) is at a relatively higher altitude compared to the lidar site (RSP, 247 m a.s.l.), as shown in Figure 1b. Additionally, the wind speed is much higher ($2.2 \pm 0.6$ m/s) for case 2, as shown in Figure S8. This higher wind speed can induce updrafts, causing an increase in the boundary layer height.

Daily evolution of PM10 and boundary layer is presented by a two-year database. The authors should explain

discrepancies between this evolution and that observed in their one-month period.

R: The correlation between boundary layer and aerosol distribution revealed in our short-term study fitted well with the 2-year dataset, which supports the robustness of our results. However, there are still some discrepancies which need to be discussed. The seasonal analysis shows that the ground-level $PM_{10}$ concentrations are correlated with boundary layer heights from 04:00 to 08:00 (UTC) for all datasets. However, the strength of the correlation is different for different seasons. The spring (MAM) shows the strongest correlation (Pearson correlation coefficient: 0.83) while the winter (DJF) shows the weakest correlation (Pearson correlation coefficient: 0.26). In addition, the summer has the highest mixing layer height (1283 $\pm$ 399 m) while the winter has the lowest mixing layer height (682 $\pm$ 542 m) as expected due to the solar radiation being strongest in summer while weakest during winter. The ground-level $PM_{10}$ aerosol concentrations are anti-correlated with mixing layer heights and show the highest concentrations during winter (33 $\pm$ 32 $\mu$g/m$^3$) and the lowest concentrations during summer (16 $\pm$ 7 $\mu$g/m$^3$).

This also shows that the ground-level $PM_{10}$ concentrations are correlated with boundary layer heights from 04:00 to 08:00 (UTC) for all datasets. However, the strength of the correlation is different for different seasons. The spring (MAM) shows the strongest correlation (Pearson correlation coefficient: 0.83) while the winter (DJF) shows the weakest correlation (Pearson correlation coefficient: 0.26). In addition, the summer has the highest mixing layer height (1283 $\pm$ 399 m) while the winter has the lowest mixing layer height (682 $\pm$ 542 m) as expected due to the solar radiation being strongest in summer while weakest during winter. The ground-level $PM_{10}$ aerosol concentrations are anti-correlated with mixing layer heights and show the highest concentrations during winter (33 $\pm$ 32 $\mu$g/m$^3$) and the lowest concentrations during summer (16 $\pm$ 7 $\mu$g/m$^3$).

The authors should indicate the reasons to select some specific days, such as February 13$^{th}$ or 14$^{th}$.

R: The two cases were chosen due to their characteristic atmospheric conditions. The case from February 13$^{rd}$ to February 14$^{th}$ was selected due to the low wind speed (0.76 $\pm$ 0.35 m/s). The low wind speed minimizes the impact of horizontal transport, allowing for more accurate analysis of local atmospheric conditions. Additionally, the clear skies during these two days ensured sufficient solar radiation to fully engage the boundary layer dynamics. In contrast, the case from February 24$^{rd}$ to February 25$^{th}$ was chosen due to the presence of clear skies but with relatively stronger wind speeds (2.2 $\pm$ 0.6 m/s). This selection allows for a comparative analysis of these two cases, highlighting the differences that wind speed can introduce to atmospheric conditions under otherwise similar solar radiation conditions.

The case from February 13$^{rd}$ to February 14$^{th}$ was selected due to the low wind speed (0.76 $\pm$ 0.35 m/s). The low wind speed minimizes the impact of horizontal transport, allowing for more accurate analysis of local atmospheric conditions. Additionally, the clear skies during these two days ensured sufficient solar radiation to fully engage the boundary layer dynamics. In contrast, the case from February 24$^{rd}$ to February 25$^{th}$ was chosen due to the presence of clear skies but with relatively stronger wind speeds (2.2 $\pm$ 0.6 m/s). This selection allows for a comparative analysis of these two cases, highlighting the differences that wind speed can introduce to atmospheric conditions under otherwise similar solar radiation conditions.

Finally, future research lines could be introduced at end of conclusions.

R: In future work, we plan to conduct more sensitive test experiments and to analyse physical and chemical processes in aerosol transformations in comparison with refined PALM-4U model results.

Regarding future work related to this study, this model version did not consider the aerosol chemical composition and only $PM_{2.5}$ and $PM_{10}$ were predicted via prognostic scalar transport equations. Hence the formation of secondary aerosol generated by chemical reactions is not considered in this work and it would be the next step of our work to include this. In scope of this study we did not have computational resources and manpower to set up and test a full SALSA aerosol physics simulation. Given that we attempted the first winter evaluation of PALM's aerosol simulation behavior in complex urban terrain, our objective was to mostly check for plausible boundary layer dynamics and spatial patterns. More research is needed with more resources to address this in detail. Also, we acknowledge that the current utilization of the model is somewhat limited and we are aware of the substantial potential of the PALM-4U model for a more detailed comparison with our comprehensive observations. For instance, we plan to conduct sensitivity tests on the impact of clouds on boundary layer evolution, on the aerosol mixing processes as well as aerosol physical and chemical transformation processes. Finally, in an upcoming study we aim to present more details of the model simulations in the context of different dynamic processes.

**Minor remarks:**

1. Figure 1. Indicate if 1 km and 10 m are the network resolutions. Moreover, magnitudes and units in scales

must be introduced.

R: We have added the grid spatial resolution in caption and the magnitudes and units in scales is shown in "x-lable" and "y-label".

Two meter temperatures (contour) and ten meter winds (vectors) from the WRF simulation over the shaded model topography height in m above sea level are shown in (a). The white labels serve for orientation and the white lines mark the approximate domain boundaries. **The "5 km" and "1 km" shown in the left-upper corner of boundaries represent grid spatial resolution**. Around Stuttgart the PALM-LES domain boundaries are shown by a small white box. In (b) the PALM-4U domains are presented using the same type of visualization for the same model output time. Shown are potential temperature and horizontal winds on the second model level above surface, (i.e., 15 meter a.g.l.). The labels indicate measurement site locations and the white line indicates the aerosol laser scan beam, while the orange line indicates the location of the vertical section evaluated from PALM-4U (RSP = Rosenstein Park). **The "40 m" and "10 m" shown in the left-upper corner of boundaries represent grid spatial resolution.**

2. L. 148.Replace "haar" by "Haar".

R: We have replaced "haar" by "Haar".

3.L. 154. Replace "Zmin" by "zmin".

R: We have replace "Zmin" by "zmin"

4. L. 250, Replace "labled" by "labeled".

R: We have replaced "labled" by "labeled".

5. L. 301. Replace "conclud" by "conclude".

R: We have replaced "conclud" by "conclude".

6. Figure 9. Labels should be introduced.

R: We have added the introduction of the labels in the caption.

Range-height cross of $PM_{2.5}$ concentrations from scanning aerosol lidar (a, c) and PALM-4U simulation (b, d) at two different periods on February $14^{th}$, 2018. (a, b: 09:12 - 09:20 **(early morning)**; c, d: 16:07-16:15 **(later afternoon)**

7. References should follow the journal style.

R: We have checked the reference carefully to follow the journal style.

8. L. 716. Replace "Fro¨hlich" by "Fröhlich".

R: We have replaced "Fro¨hlich" by "Fröhlich"